# UEval: A Real-World Benchmark for Unified Multimodal Generation

## Abstract

We introduce UEval, a challenging real-world benchmark for multimodal generation of *unified models*, *i.e.*, models capable of generating both images and text. UEval comprises 1,000 expert-curated prompts that require both images and text in the model outputs, sourced from 8 diverse real-world domains. Our curated questions cover a wide range of reasoning types, from step-by-step guides to textbook explanations. Evaluating open-ended multimodal generation is non-trivial, as simple LLM-as-a-judge methods can miss the subtleties. To address this, we design a rubric-based scoring system in UEval: reference images and text are provided as inputs, an LLM generates an initial rubric for each question, and human experts refine it to ensure reliability. This question-specific rubric design allows for more tailored and accurate assessment. UEval is designed to be highly challenging: GPT-5-Thinking scores only 61.7 out of 100, while the best open-source model reaches merely 27.1. We observe reasoning models consistently outperform non-reasoning ones, and transferring reasoning traces from a reasoning model to a non-reasoning model significantly narrows the gap. This suggests that "reasoning" may be essential for tasks requiring complex multimodal understanding and generation. The dataset, code, and results will be publicly released.

## 1 Introduction

Unified models (Tong et al., 2024b; Zhou et al., 2025a; Deng et al., 2025) aim to integrate multimodal understanding and generation within a single system. Current evaluations of these models remain largely confined to visual question answering (Marino et al., 2019; Liu et al., 2024b; Yue et al., 2024; Fu et al., 2025), which requires generating a textual answer from an input image, and text-to-image generation (Huang et al., 2023; Ghosh et al., 2023; Lin et al., 2024), which takes a textual description as input and asks the model to produce a corresponding image.

As demonstrated in Figure 1, these paradigms overlook a central component of human multimodal reasoning: unified multimodal generation that produces both text and images in response to a single, complex query. Without such evaluation, current benchmarks fail to capture the rich interplay between language and vision that characterizes real-world multimodal reasoning.

While recent efforts (An et al., 2023; Liu et al., 2024a; Xia et al., 2024; Niu et al., 2025; Zhao et al., 2025) have proposed new benchmarks to evaluate unified models, there remains a lack of standardized approaches to evaluate unified multimodal generation. To address this unmet need, we introduce UEval, a simple yet challenging benchmark to assess unified models (Wang et al., 2024; Chen et al., 2025b; Yang et al., 2025; Google, 2025b; Xie et al., 2025) at scale. Unlike prior benchmarks, UEval requires models to reason and respond to complex user queries jointly in images and natural language, providing a rigorous testbed across diverse real-world scenarios.

UEval comprises 1,000 expert-curated questions spanning eight diverse real-world scenarios—*space*, *textbook*, *paper*, *diagram*, *art*, *life*, *tech*, and *exercise*. For consistent and reproducible evaluation, we propose a rubric-based framework inspired by Arora et al. (2025) since evaluating through simple "LLM-as-a-judge" is inadequate. For each question, experts first manually collect the ground-truth reference answers in both text and image. Then, a frontier multimodal Large Language Model generates an initial rubric conditioned on the original query and reference answer. Human annotators further refine these rubrics to eliminate redundancies and add any missing criteria. In total, UEval contributes 11.3K rigorously validated rubric criteria, enabling reliable automatic

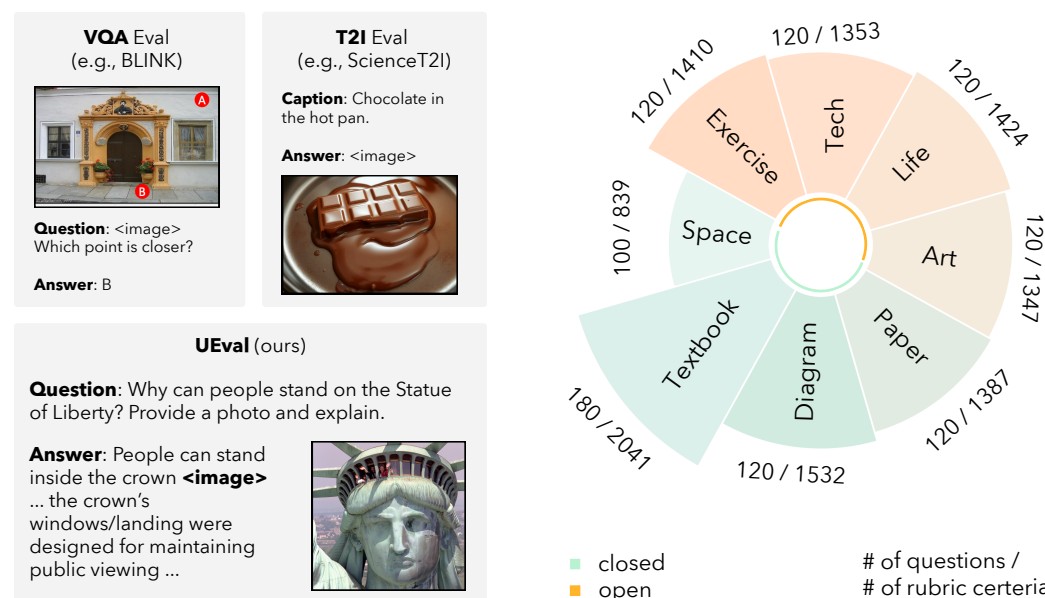

Figure 1: **Left**: Previous unified model evaluations focus on either image understanding (*i.e.*, VQA) or image generation from captions (*i.e.*, T2I). In contrast, UEVAL requires models to reason across modalities and generate responses in both images and text. The VQA example is from BLINK (Fu et al., 2024), and the T2I example is from ScienceT2I (Li et al., 2025a). **Right**: The chart illustrates the number of questions and corresponding rubric criteria across tasks in UEVAL.

grading. In our experiments, we employ Gemini-2.5-Pro (Google, 2025a) as a rubric-based judge to score model responses, given that its evaluations exhibit strong agreement with human judgments.

We conduct a comprehensive evaluation of 8 unified models on UEVAL. Results reveal that our benchmark presents a great challenge to all models. Among them, GPT-5-Thinking (OpenAI, 2025) achieves the highest score of 61.7 out of 100 averaged across all tasks, whereas the best-performing open-source model (*i.e.*, BAGEL (Deng et al., 2025)) reaches only 27.1 out of 100. The gap between average open-source and proprietary scores is as large as 38.7. In addition, our experiments show current unified models struggle to generate multiple images with consistent labeling over time.

Interestingly, we observe a systematic advantage of "reasoning" models (*e.g.*, GPT-5-Thinking) over their non-reasoning counterparts (*e.g.*, GPT-5-Instant). To further investigate the benefit of explicit reasoning in multimodal generation, we consider a simple experiment: we append the reasoning trace produced by GPT-5-Thinking to the end of the original question prompt and feed it into non-reasoning models. Surprisingly, this substantially improves visual outputs generated by GPT-5-Instant and Gemini-2.5-Flash, while open-source models (*e.g.*, BAGEL) show no improvement. These observations suggest that chain-of-thought reasoning (Wei et al., 2022), long studied in text-only LLMs, may also play an important role in enabling unified multimodal generation.

## 2 UEVAL

We present UEVAL, a benchmark specifically designed to evaluate *multimodal* generation. UEVAL focuses on real-world tasks where a model must reason carefully before generating natural language and images in response to user queries. UEVAL consists of 8 carefully designed tasks in the real world. As illustrated in Figures 2 and 3, these tasks range from explanations that rely on visual illustrations (*e.g.*, space) to academic figure creation (*e.g.*, paper). They also vary in format, from multi-step generation to single-step description (guide *vs*. machine learning concept) and in breadth, from very general content to highly specialized topics (textbook *vs*. paper).

Each question in our benchmark includes a text prompt and a grading rubric for evaluating model outputs. For all of the tasks, we also provide reference images and texts sourced from diverse

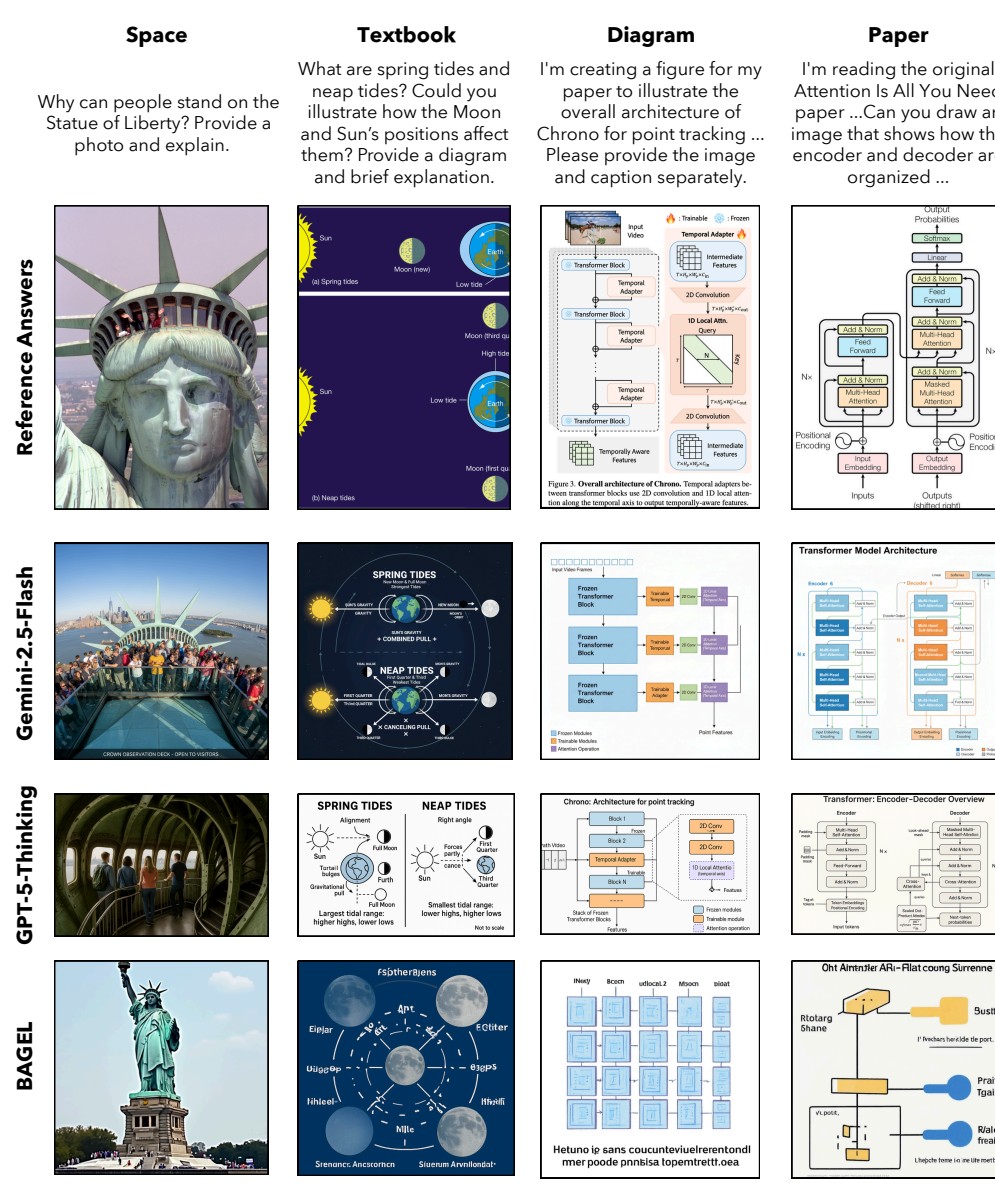

Figure 2: **Generated images on UEVAL tasks with reference answers**. We visualize images generated by GPT-5-Thinking, Gemini-2.5-Flash, and BAGEL (Deng et al., 2025). These images fail to answer the questions accurately. For example, Gemini-2.5-Flash depicts a nonexistent external platform above the Statue of Liberty instead of the actual interior one inside the crown.

materials (*e.g.*, Wikipedia, textbooks, social media, and existing datasets). Our evaluation rubrics are drafted by Gemini-2.5-Pro initially and then refined and verified by a human (details in Section 2.2). As shown in Figure 1, UEVAL contains 1,000 questions and 11,333 rubric criteria in total.

## 2.1 DATASET COMPOSITION

We describe each task in UEVAL below. We group the tasks *art*, *life*, *tech*, and *exercise* under a broader category *guide* as they share similar design principles. Additional details about the image sources and text sources of our benchmark are provided in Appendix B.

**Space**. This task evaluates a model's ability to depict special architecture or engineering features. Crucially, the generated images must highlight the structural elements relevant to the question rather than serve as decoration. For example, given the prompt *"Why can people stand on the Statue of*

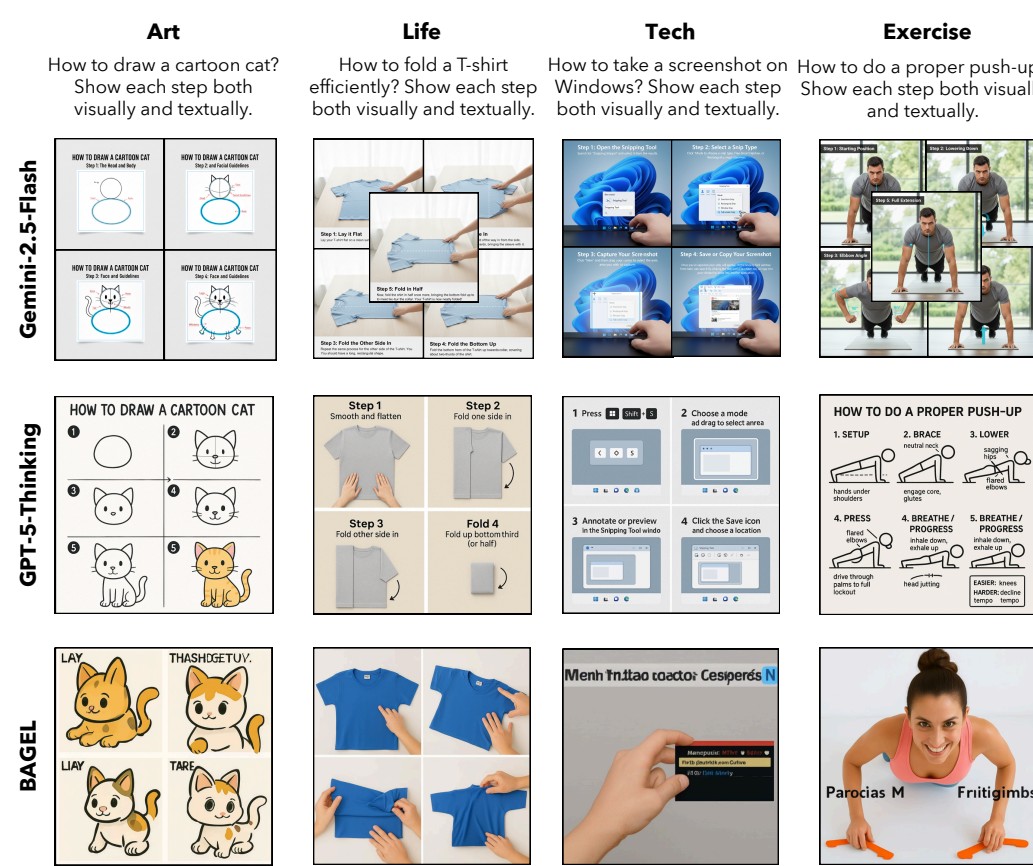

Figure 3: **Generated images on UEVAL tasks without reference answers**. We prompt GPT-5-Thinking, Gemini-2.5-Flash, and BAGEL to synthesize step-by-step visual guides for each task. The generated images often exhibit temporal inconsistencies. For instance, in the *art* task, when drawing a cartoon car, GPT-5-Thinking mislabels sub-images (*e.g.*, two images both tagged as step 5). For visualization, we stack the Gemini-2.5-Flash results into a single grid.

*Liberty? Provide a photo and explain.",* a full-view photo of the statue would be incorrect because it fails to reveal the crown platform where visitors can stand.

To construct this task, We first collect a small set of 20 seed questions about well-known real-world landmarks from online Q&A forums (*e.g.*, Quora-like platforms). These questions focus on the engineering or structural features of a landmark. Since there are not many questions available across the Internet and in public datasets, we use GPT-5 to expand this set. Given our seed questions, the model is prompted to propose additional ones in the same style but targeting different landmarks. Human annotators then review these questions to ensure that each one refers to an actual landmark.

For every verified question, annotators retrieve a representative real image from public sources (*e.g.*, Wikipedia) that depicts the relevant architectural feature and answers the question visually. Finally, annotators write a short reference text describing how the chosen image illustrates the underlying engineering property for that landmark.

**Textbook**. This task tests a model's ability to explain fundamental science phenomena (*e.g.*, geological transformations or chemical reactions) through instructional diagrams and captions. The generated outputs should help reveal underlying mechanisms (*e.g.*, DNA's role in genetics) or connections between concepts (*e.g.*, how a headland erodes into caves, arches, and stacks). An example question is *"I don't quite get how a headland turns into caves, arches, and stacks. Can you generate an image and explain the sequence? Please answer with both visual and textual explanations."*

We use the textbook-style diagrams and their corresponding answer texts from the TQA dataset (Kembhavi et al., 2017) as reference image–text pairs. These pairs cover several core subject

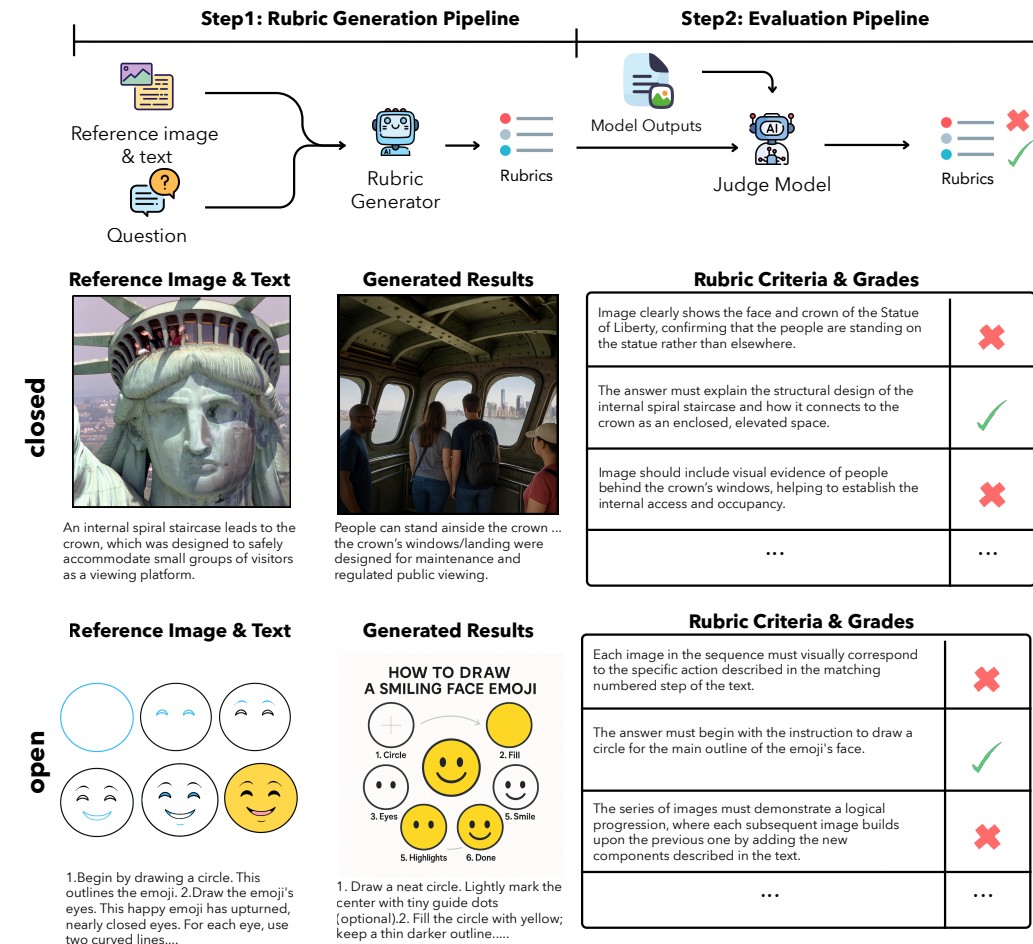

Figure 4: **Rubric drafting and response evaluation procedure in UEVAL**. We propose using *data-dependent* rubrics to evaluate outputs from unified multimodal LLMs. For each question, a model drafts an itemized rubric based on the question and the reference image-text pair (if provided). A judge model then scores the generated response against each rubric criterion to do the evaluation.

areas in middle-school science education, including biology, geography, and chemistry, providing a sufficiently diverse and representative foundation for constructing our samples. Because TQA does not supply question text that fits our task design, we do not use the question from TQA. Building on these references, we then prompt GPT-5 to generate learning-oriented questions. Human annotators then manually review the generated questions to check each one is scientifically appropriate, unambiguous, and fully consistent with reference answers. This process yields a collection of high-quality textbook-style questions, each aligned with a specific diagram and corresponding explanatory text.

**Diagram**. This task targets real-world scenarios in academic writing, where researchers must create figures that illustrate complex methods in research papers. The evaluated model receives a technical description of a specific framework or model architecture and needs to synthesize a self-contained figure accompanied by an informative caption.

To construct this task, we manually curate figures from recent top-tier AI conference papers (*e.g.*, ICLR, CVPR), and use each figure with its original caption as the reference image–text pair. We intentionally avoid well-known canonical diagrams to prevent memorization. For each figure, we provide GPT-5 with the source paper to draft a generation instruction. Human annotators then review and refine the prompt–caption pair to ensure conceptual correctness, faithfulness to the original scientific intent, and independence from external context. This process yields high-quality diagram–caption samples aligned with real research methods.

**Paper**. This task assesses whether a unified model can clearly explain complex concepts from cutting-edge computer science research in a clear and accessible manner. Given a high-level user question about a particular method or architecture, the model must first understand the technical content and then address the user's question in an accessible way. An example question is *"I am reading the Transformer paper—can you draw and explain the encoder–decoder structure?"*

To build this task, We source a diverse set of reference figures from seminal papers (*e.g.*, Transformer (Vaswani et al., 2017), ResNet (He et al., 2016)) and online teaching platforms (*e.g.*, D2L (Zhang et al., 2023)). These figures capture key innovations in modern machine learning, ranging from neural network architectures to optimization pipelines. For each selected figure, we extract the corresponding technical descriptions from the source material and provide them to GPT-5, prompting it to generate natural reader-oriented questions along with detailed reference answers that unpack the central concepts illustrated by the figure. Human annotators then review the generated questions and answers to ensure conceptual accuracy, faithfulness to the original research.

**Guide**. This evaluates a model's ability to produce a coherent, step-by-step visual guide for everyday activities. It contains four tasks—*art*, *life*, *tech*, and *exercise*–to cover a broad range of real-world skills that require multi-step demonstration. For each question, the model must generate a visual guide (in one or more images) together with accompanying textual instructions that jointly illustrate a clear progression from the initial state to the final outcome. Although illustrations at consecutive steps may differ only slightly, the model must preserve temporal consistency across the sequence of images.

Questions are collected from high-quality tutorials (*e.g.*, wikiHow, Easy Drawing Guides) and demonstrational videos (*e.g.*, YouTube). Questions are sourced from high-quality tutorial materials (e.g., wikiHow, Easy Drawing Guides) and demonstrational videos (e.g., YouTube). Each question is accompanied by a reference answer to support our data-dependent evaluation process. When constructing the rubrics, we design them to focus on generalizable, task-agnostic evaluation dimensions—such as step structure, text–image alignment, and temporal coherence—rather than enforcing a single, content-specific correct solution. For example, in the *art* task, for the question *"How to draw a cat?"*, the rubric evaluates structural and procedural properties of the output rather than the exact drawing style. It includes criteria such as: *"The answer must describe a step-by-step process for drawing a cat."* and *"The sequence of images must illustrate the complete drawing process, from the initial basic shapes to the final version."*

## 2.2 Evaluation with Rubrics

How to effectively evaluate unified multimodal generation remains an open problem. Zhou et al. (2025b) use average win rates from pairwise comparisons of model outputs containing images and text, but this approach requires a large number of comparisons to get accurate model scores, and the scores can change if a different set of models is evaluated. Moreover, most current evaluations (Xia et al., 2024; Zhou et al., 2025b) are *data-independent*: a single generic prompt is applied to grade all samples. As a result, this overlooks sample-specific differences and can lead to inaccurate results.

**Rubric generation**. Inspired by HealthBench (Arora et al., 2025), we adopt a *data-dependent* approach to evaluate unified multimodal generation. Figure 4 (*top*) illustrates our framework to create rubrics with an multimodal Large Language Model based on the question and reference answers. Since all tasks in our benchmark include reference pairs, we construct two itemized rubrics—one for image and one for text—using both the question and its reference pair.

**Human review**. For each benchmark sample, a primary annotator performs the first-round verification, including the question, the reference answer, and an initial rubric. Other co-authors then independently review the annotation. Human annotators therefore supervise benchmark construction at a system level, including the design of inputs (questions), outputs (reference answers), and grading criteria (rubrics). Our objective is to design rubrics that reward the reference answer while appropriately penalizing incorrect or misaligned model outputs. Only rubric items unanimously judged by all reviewers to be clear and well aligned with the task design are retained.

Human annotators apply several types of modifications and quality-control checks to the model-generated rubric items. First, redundant or overlapping criteria are removed or consolidated—for example, merging *"the steps must be sequential"* and *"the steps should follow a logical order"*

| | Space | Textbook | Diagram | Paper | Art | Life | Tech | Exercise | Avg |
|---|---|---|---|---|---|---|---|---|---|
| | | | *Open-source Models* | | | | | | |
| Janus-Pro | 18.6 | 26.0 | 33.8 | 14.5 | 19.7 | **15.4** | 14.3 | 10.8 | 19.1 |
| Show-o2 | 22.0 | 31.3 | 28.7 | 16.9 | 17.4 | 11.4 | 11.5 | 11.9 | 18.9 |
| MMaDA | 9.5 | 17.5 | 13.5 | 6.4 | 10.9 | 10.3 | 9.2 | 18.8 | 12.0 |
| BAGEL | **29.8** | **42.5** | **37.2** | **20.0** | **31.1** | 12.8 | **21.6** | **21.9** | **27.1** |
| | | | *Proprietary Frontier Models* | | | | | | |
| Gemini-2.0-Flash | 62.8 | 56.2 | 44.1 | 44.4 | **58.4** | 43.2 | 41.4 | 41.6 | 49.0 |
| Gemini-2.5-Flash | 76.4 | 72.8 | **62.2** | **70.4** | 53.4 | 55.8 | **56.1** | 46.0 | 61.6 |
| GPT-5-Instant | 76.8 | **76.3** | 58.2 | 63.4 | 53.4 | **57.0** | 42.8 | 50.2 | 59.7 |
| GPT-5-Thinking | **83.6** | 76.1 | 62.1 | 53.0 | 57.8 | 54.5 | 51.1 | **55.6** | **61.7** |

Table 1: **UEVAL Leaderboard**. We evaluate both open-source and proprietary frontier models on 8 tasks in UEVAL. The best performance for each task and overall is in bold.

into a single, more precise item. Second, missing but essential evaluation dimensions are added, such as *"All visible text in the image(s) must be spelled correctly and rendered naturally, with no misspellings, garbled characters, distortions, or nonsensical text."* when readability is relevant. Third, annotators ensure that all rubric items are unambiguous and capable of reliably distinguishing correct outputs from incorrect or misaligned ones.

**Rubric Evaluation**. We finally use an external multimodal Large Language Model (*e.g.*, Gemini-2.5-Pro) to judge which rubric items each model response (*i.e.*, generated images and text) satisfies, and compute the final score as the fraction of satisfied items over the total number of rubric criteria in a test sample. This provides an automated grading method to replace human evaluation. We find using a frontier model yields results well aligned with human judgments and will discuss this in the next section.

## 3 EXPERIMENTS

### 3.1 SETTINGS

**Models**. We evaluate recent unified models on all 8 tasks in our benchmark. For open-source models, we consider Janus-Pro-7B (Chen et al., 2025c), Show-o2 (Xie et al., 2025), MMaDA (Yang et al., 2025), and BAGEL (Deng et al., 2025). For proprietary frontier models, we evaluate GPT-5-Instant (OpenAI, 2025), GPT-5-Thinking (OpenAI, 2025), Gemini-2.0-Flash (Google, 2024), and Gemini-2.5-Flash (*a.k.a.*, Nano Banana) (Google, 2025b).

**Evaluation setup**. Some models (*e.g.*, MMaDA, BAGEL) can generate only images or texts by design, but not both in a single inference pass. To obtain both outputs, we feed the same text prompt to the model twice—once for the image and once for the text. For models that natively support joint image-text generation (*e.g.*, GPT-5, Gemini), we directly collect their multimodal responses.

To score a model's response, we use Gemini-2.5-Pro (Google, 2025a) to grade the generated content based on fine-grained rubrics (Section 2.2). We observe that the scores are largely consistent with human evaluations. More details on evaluation (*e.g.*, the prompts used) can be found in Appendix C. More quantitative comparisons are provided in Section 3.3.

### 3.2 RESULTS

Table 1 reports the performance of various models on UEVAL. Overall, frontier models consistently outperform open-source ones across all tasks. Among them, GPT-5-Thinking achieves the highest average score of 61.7 out of 100, while the best open-source model obtains only 27.1. To understand performance differences more qualitatively, Figure 6 presents a radar chart comparing models across tasks. The gap between proprietary and open-source models is strikingly large: the strongest frontier model (*i.e.*, GPT-5-Thinking) scores more than twice as high as the best-performing open-source model (*i.e.*, BAGEL), almost on every task. The individual image and text scores for each task are provided in Appendix A.

**Question**: Why can people stand on the Statue of Liberty? Provide a photo and explain.

**Reference Answer**

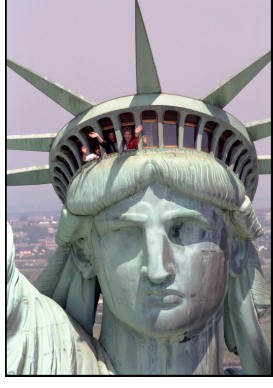

**GPT-5-Instant**   **Gemini-2.5-Flash**   **BAGEL**

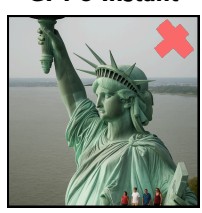   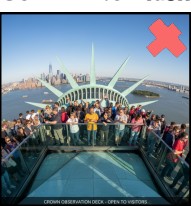   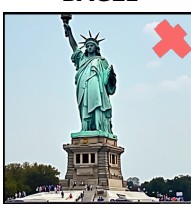

⬇ **After Adding Reasoning Trace by GPT-5-Thinking**: Only the crown is open to visitors (the torch has been closed since 1916), and reaching the crown requires ... The crown contains a circular room with windows. Visitors' weight is transferred through brackets and the spiral ... The pedestal is made of reinforced concrete and granite. The statue is anchored by large iron tie bars to the ...

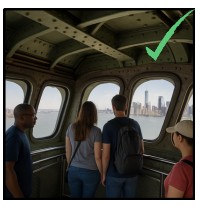   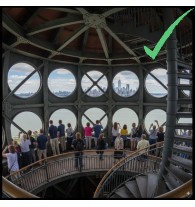   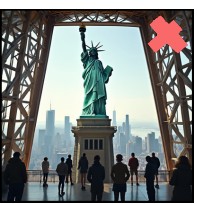

Figure 5: **Reasoning can improve the quality of multimodal generation**. We extract the *reasoning trace* from GPT-5-Thinking and append it to the end of the original question. The modified prompt is then fed into *non-reasoning* models (*e.g.*, GPT-5-Instant, Gemini-2.5-Flash, and BAGEL) to generate responses. This can push the generated image toward that of the reasoning model and yield a more accurate response. Note BAGEL still generates a full-view image of the Statue of Liberty.

We also observe that the tasks requiring multi-step planning (*e.g.*, art, life) yield substantially lower scores than knowledge-based tasks (*e.g.*, textbook, diagram). Figure 3 further illustrates this pattern. For example, in the *art* task, GPT-5-Thinking incorrectly labels the final two images as *step 5*. Similar mistakes also occur in both *life* and *exercise* tasks. Likewise, Gemini-2.5-Flash changes the cloth's orientation from step 1 to step 2 and then reverts it in step 3 in the *life* task.

More interestingly, the current *reasoning* model (*e.g.*, GPT-5-Thinking) achieves significantly better performance than the non-reasoning one (*e.g.*, GPT-5-Instant). We further examine the benefits of *reasoning* in multimodal generation in the next section.

## 3.3 ANALYSIS

**Image and text generation quality**. As described in Section 2.2, we create two separate rubrics for each question to evaluate image and text quality independently. The individual image and text scores are reported in Appendix A. Overall, models achieve higher text scores than image scores. Open-source models, in particular, perform reasonably well on text generation but extremely poorly on image generation, often receiving only single-digit image scores. By contrast, frontier models exhibit a much smaller gap between their image and text generation abilities, yet still show noticeable degradation on image-based criteria, highlighting that faithful visual generation remains significantly more challenging than text generation.

**The effectiveness of *reasoning traces***. To understand why reasoning models (*e.g.*, GPT-5-Thinking) yield better results in multimodal generation, we record a reasoning trace and append it to original question prompt. We then provide non-reasoning models (*e.g.*, GPT-5-Instant) with this modified prompt. Figure 5 presents the results. Surprisingly, incorporating the reasoning trace enables GPT-5-Instant and Gemini-2.5-Flash to generate a more accurate image of the interior of the Statue of Liberty's crown. This suggests that the multimodal generation of unified models can benefit from Chain-of-Thought (Wei et al., 2022) reasoning, even when it is generated by other models.

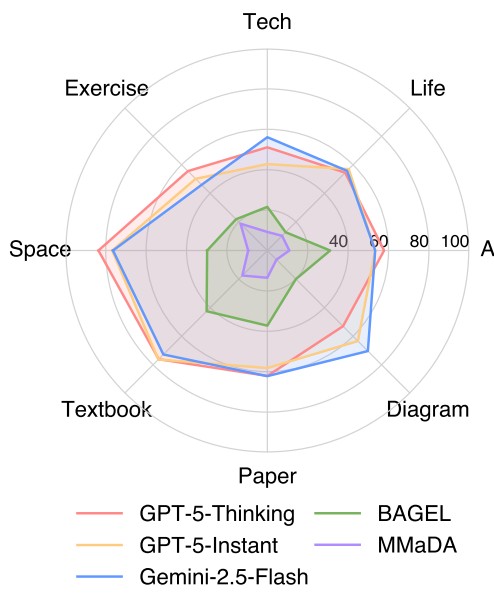

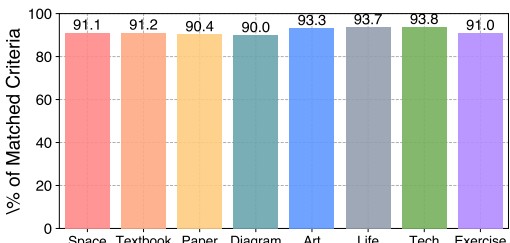

Figure 7: Percentage of matched rubric criteria between human evaluator and an LLM judge

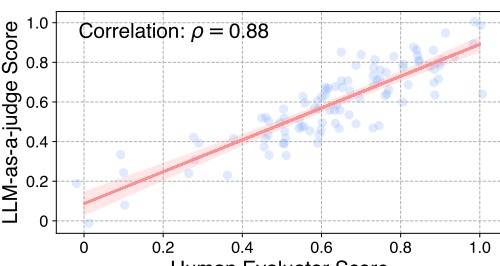

Figure 6: Accuracies of models on UEVAL

Figure 8: LLM-as-a-Judge *vs.* Human Evaluator

**Human evaluation**. To assess the reliability of using an LLM as a judge, we conduct a human evaluation of model responses based on our rubric criteria (Section 2.2). Specifically, we randomly sample 10% of the questions from each task and ask human annotators to determine how many rubric criteria each model response satisfies. In Figure 7, we report the percentage of matched rubrics between human evaluators and LLM-as-a-judge. Most LLM-as-a-judge gradings align closely with human judgment. Moreover, in Figure 8, we plot the LLM-as-a-judge scores *vs.* human evaluator scores for each question and report a strong Pearson correlation of *0.88* between them. This indicates the robustness and reliability of our automated evaluation framework.

**Different judge models**. To further ensure the robustness and reproducibility of our evaluation framework, we additionally assess rubric–response matching using a diverse set of open-weight and frontier multimodal LLMs as judges. In our experiments, we used the following open-weights models and frontier models: Qwen3-VL-8B (Bai et al., 2025), InternVL-3.5-8B (Wang et al., 2025), GLM-4.1V-9B-Thinking (Team et al., 2025), Qwen3-VL-235B (Bai et al., 2025), Doubao-Seed-1.6Vision (ByteDance, 2025), and GPT-5 (OpenAI, 2025) on GPT5-thinking results. Table 2 reports the resulting per-task average scores under these different judges.

We observe that the strongest judges—GPT-5 and Gemini-2.5-Pro—produce highly consistent scores across all tasks. In contrast, current open-weights multimodal models (e.g. Qwen3-VL-8B) remain considerably limited as judges. Our findings show that while open-weights models are valuable for reproducibility, they exhibit substantially higher score variance across tasks. Therefore, we recommend using a stable snapshot of a frontier API model as the primary judge. Open-weights models can serve as supplementary evaluators, but they currently cannot replace Gemini-2.5-Pro in this evaluation pipeline without compromising the benchmark's stability and reliability.

## 4 RELATED WORK

**Multimodal Large Language Models and unified models**. Multimodal Large Language Models (MLLMs) have progressed greatly in recent years. These models (Alayrac et al., 2022; Dai et al., 2023; Liu et al., 2023) typically integrate a visual encoder (Radford et al., 2021; Dosovitskiy et al., 2021) with a pre-trained Large Language Model (Llama-2-Team, 2023; Peng et al., 2023), achieving strong performance on image captioning (Chen et al., 2015; Plummer et al., 2015) and visual question answering (Goyal et al., 2017; Mathew et al., 2021). To further scale their capabilities, these

| | Space | Textbook | Diagram | Paper | Art | Life | Tech | Exercise |
|---|---|---|---|---|---|---|---|---|
| Gemini-2.5-Pro | 83.6 | 76.2 | 62.1 | 53.0 | 57.8 | 54.6 | 51.1 | 55.6 |
| GPT-5 | 80.0 | 73.0 | 55.8 | 46.3 | 56.4 | 50.1 | 45.4 | 51.7 |
| Qwen3-VL-8B | 82.2 | 85.3 | 75.4 | 55.4 | 84.4 | 79.3 | 78.2 | 76.8 |
| InternVL-3.5-8B | 79.9 | 87.6 | 68.2 | 53.6 | 68.8 | 72.4 | 71.4 | 72.6 |
| GLM-4.1-Thinking | 84.3 | 83.6 | 68.2 | 49.8 | 79.4 | 74.5 | 74.3 | 70.7 |
| Qwen3-VL-235B | 82.0 | 85.2 | 72.3 | 53.6 | 73.7 | 61.4 | 57.7 | 63.0 |
| Doubao-Seed-1.6Vision | 85.5 | 81.0 | 68.2 | 53.8 | 75.2 | 70.8 | 67.8 | 70.9 |

Table 2: Average scores for each task across different judge models.

models need millions of instruction tuning data (Tong et al., 2024a; Deitke et al., 2025). Table **??** reports the resulting per-task average scores under these different judges.

A parallel line of research seeks to unify multimodal understanding and generation within a single MLLM. Some methods adopt diffusion-based methods (Dong et al., 2024; Yang et al., 2025; Li et al., 2025c), whereas others train models purely with an auto-regressive objective (Team, 2024; Wu et al., 2025a; Wang et al., 2024; Wu et al., 2025b). There are also hybrid methods that combine both approaches (Deng et al., 2025; Zhou et al., 2025a; Xie et al., 2025; Chen et al., 2025a). We refer readers to Zhang et al. (2025) for a comprehensive survey of MLLMs and unified models.

**Multimodal benchmarks**. A range of benchmarks has been proposed to evaluate multimodal inputs. Initial work (Goyal et al., 2017; Marino et al., 2019; Masry et al., 2023) evaluate image understanding for specific image types, and later efforts benchmark on broader image coverage (Liu et al., 2024b; Yu et al., 2024; Fu et al., 2025). There are also studies evaluating text-to-image generation quality (Huang et al., 2023; Ghosh et al., 2023; Lin et al., 2024). Some benchmarks (*e.g.*, VDC (Chai et al., 2025)) start to use a data-dependent rubric for better evaluation. More recent works unify both understanding and generation benchmarks as interleaved text-and-image generation (An et al., 2023; Liu et al., 2024a; Chen et al., 2024) or unified multimodal generation (Li et al., 2025b).

Different from them, our benchmark targets real-world tasks that require models to reason and respond to questions in both natural language and images. Moreover, our evaluation is very simple: an MLLM judge is used to grade model responses based on data-dependent rubrics. This eliminates the need for human evaluation (Zhou et al., 2025b) or training a scoring model (Xia et al., 2024).

## 5 LIMITATION AND FUTURE WORK

While our benchmark is designed to evaluate unified models in real-world scenarios, its current scale is relatively small and may not comprehensively capture the full spectrum of real-world diversity. In addition, both our rubric drafting and grading framework rely on an external LLM to generate and evaluate the outputs from other models. This might introduce model bias (*i.e.*, potentially favoring some evaluated responses over others (Xu et al., 2024; Wataoka et al., 2024)).

The current version of our benchmark does not contain any questions that use an image as input for multimodal generation. This may overlook the cases in which unified models receive visual information in the prompt and respond to the question with an image as well. In future work, we will incorporate such tasks that require generation based on input images (*e.g.*, diagram completion).

## 6 CONCLUSION

We introduce UEVAL, a benchmark to evaluate unified multimodal generation beyond standard tasks (*e.g.*, visual question answering or text-to-image generation). Our benchmark contains 1,000 samples across 8 real-world tasks and provides 11,333 fine-grained rubric criteria for rigorous, automated grading of model responses. Our results demonstrate that UEVAL is very challenging to both proprietary frontier and open-source unified models. We hope this work will stimulate further research on developing stronger unified models and better benchmarks.

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

# APPENDIX

## A INDIVIDUAL IMAGE AND TEXT SCORES

| | Space | | Textbook | | Diagram | | Paper | | Avg | |
|---|---|---|---|---|---|---|---|---|---|---|
| | Image | Text | Image | Text | Image | Text | Image | Text | Image | Text |
| Reference | 91.4 | 92.5 | 92.5 | 96.2 | 90.4 | 90.5 | 92.2 | 95.1 | 91.6 | 93.6 |
| *Open-source Models* | | | | | | | | | | |
| Janus-Pro | 10.1 | 27.1 | 7.6 | 44.4 | 1.2 | 66.4 | 4.1 | 24.9 | 5.7 | 40.7 |
| Show-o2 | 18.0 | 26.0 | 9.9 | 52.6 | 4.8 | 52.7 | **5.4** | 28.4 | 9.5 | 39.9 |
| MMaDA | 14.2 | 4.8 | **16.6** | 18.5 | **8.4** | 18.6 | 3.4 | 9.4 | 10.6 | 12.8 |
| BAGEL | **27.6** | **31.9** | 14.5 | **70.6** | 2.8 | **71.7** | 4.8 | **35.2** | 12.4 | **52.3** |
| *Proprietary Frontier Models* | | | | | | | | | | |
| Gemini-2.0-Flash | 58.3 | 67.2 | 31.0 | 81.4 | 19.5 | 68.8 | 16.7 | 72.2 | 31.4 | 72.4 |
| Gemini-2.5-Flash | 72.8 | 79.9 | 59.4 | 86.2 | **51.1** | 73.3 | **58.8** | **81.9** | **60.5** | 80.3 |
| GPT-5-Instant | 74.0 | 79.5 | **64.2** | 88.3 | 39.2 | 77.1 | 45.6 | 81.2 | 55.8 | **81.5** |
| GPT-5-Thinking | **81.6** | **85.5** | 62.1 | **90.2** | 44.9 | **79.3** | 43.3 | 62.7 | 58.0 | 79.4 |

Table 3: **Image and text scores on closed-ended tasks in UEVAL..** We observe that models achieve much worse performance on the *image* scores of the benchmark. Reference denotes the evaluation score on reference images and texts. These values are only about 90% as the LLM judge can still make minor errors during evaluation when provided with fully correct rubrics.

| | Art | | Life | | Tech | | Exercise | | Avg | |
|---|---|---|---|---|---|---|---|---|---|---|
| | Image | Text | Image | Text | Image | Text | Image | Text | Image | Text |
| Reference | 78.6 | 95.8 | 74.0 | 98.0 | 79.8 | 92.7 | 74.1 | 98.4 | 76.6 | 96.2 |
| *Open-source Models* | | | | | | | | | | |
| Janus-Pro | 6.1 | 33.3 | 7.0 | **23.9** | 3.7 | 24.9 | 7.3 | 14.3 | 6.0 | 24.1 |
| Show-o2 | 3.6 | 31.2 | 4.4 | 18.3 | 3.7 | 19.4 | 7.0 | 16.8 | 4.7 | 21.4 |
| MMaDA | **15.9** | 5.8 | **16.4** | 4.2 | **15.0** | 3.5 | **20.6** | 17.1 | **17.0** | 7.6 |
| BAGEL | 13.4 | **48.9** | 8.8 | 16.8 | 3.8 | **39.4** | 17.3 | **26.4** | 10.8 | **32.9** |
| *Proprietary Frontier Models* | | | | | | | | | | |
| Gemini-2.0-Flash | **36.8** | 80.0 | 30.4 | 55.9 | 21.7 | 61.2 | 33.5 | 49.7 | 30.6 | 61.7 |
| Gemini-2.5-Flash | 36.3 | 70.6 | **54.5** | 57.1 | **49.0** | 63.2 | 34.7 | 57.2 | **43.6** | 62.0 |
| GPT-5-Instant | 34.9 | 71.8 | 50.4 | 63.6 | 25.9 | 59.6 | **49.3** | 51.1 | 40.1 | 61.5 |
| GPT-5-Thinking | 23.8 | **91.9** | 30.7 | **78.4** | 18.9 | **83.3** | 33.1 | **78.0** | 26.6 | **82.9** |

Table 4: **Image and text scores on open-ended tasks in UEVAL** . We observe that models achieve much worse performance on the *image* scores of the benchmark. Reference denotes the evaluation score on reference images and texts.These values range from 70–100%, as the reference answers contain multiple images and the LLM judge can still make minor evaluation errors even when provided with fully correct rubrics.

## B DATA SOURCES

We list data source for every task of our benchmark in Section 2.1.

## C EVALUATION DETAILS

Figure 9 shows the LLM-as-a-judge prompt to grade each rubric item in our evaluation.

| Task | Source | Usage |
|------|--------|-------|
| Space | Wikipedia, TripAdvisor, Google search, Quora | Question + Reference Image & Text |
| Textbook | TQA dataset | Reference Image & Text |
| Diagram | arXiv, ICLR, CVPR, NeurIPS, ICCV | Reference Image & Text |
| Paper | arXiv, d2I, medium, CS231n course website | Reference Image & Text |
| Art | WikiHow, EasyDrawing, YouTube, ArtForKidsHub | Question |
| Life | WikiHow, YouTube | Question |
| Tech | WikiHow, Food52, YouTube | Question |
| Exercise | WikiHow, Healthline, Men's Health, YouTube | Question |

Table 5: Per-task breakdown of data sources and data types in UEVAL.

**Prompting Gemini-2.5-pro to judge**

Your job is to look at a conversation and a rubric item, and score the answer in the conversation on how well it follows the rubric item. The conversation contains a list of images, a text answer, and a question. The images are to help answer the question.

**Conversation**
Question: <<question>>
Text Answer: <<text_answer>>
Image Answer: <<image_answer>>

**Rubric Item**
<<rubric_item>>

**Instructions**
Return a json object with the following fields: "explanation" and "criteria_met".
- The "explanation" field should be a string explaining why the image does or does not meet the criteria of the rubric item.
- The "criteria_met" field should be a boolean indicating whether the image meets the criteria of the rubric item. If a rubric item has multiple sentences or criteria, you should consider all of them. If any of the criteria is not met, the answer should be false. Only return true is all of the criteria are met. If there is not enough information in the image or you are not sure, the answer should be "not sure".
- If the rubric item is just about the image or diagram, you should only analyze the image, not the text answer. If the rubric item is about the image and the text answer, you should analyze both the image and the text answer.

Figure 9: Evaluation prompt used for Gemini-2.5-Pro as the judge model.

# D GENERATED RESPONSES ON MORE MODELS

In Figures 10 and 11, we visualize more examples of images generated by different models.

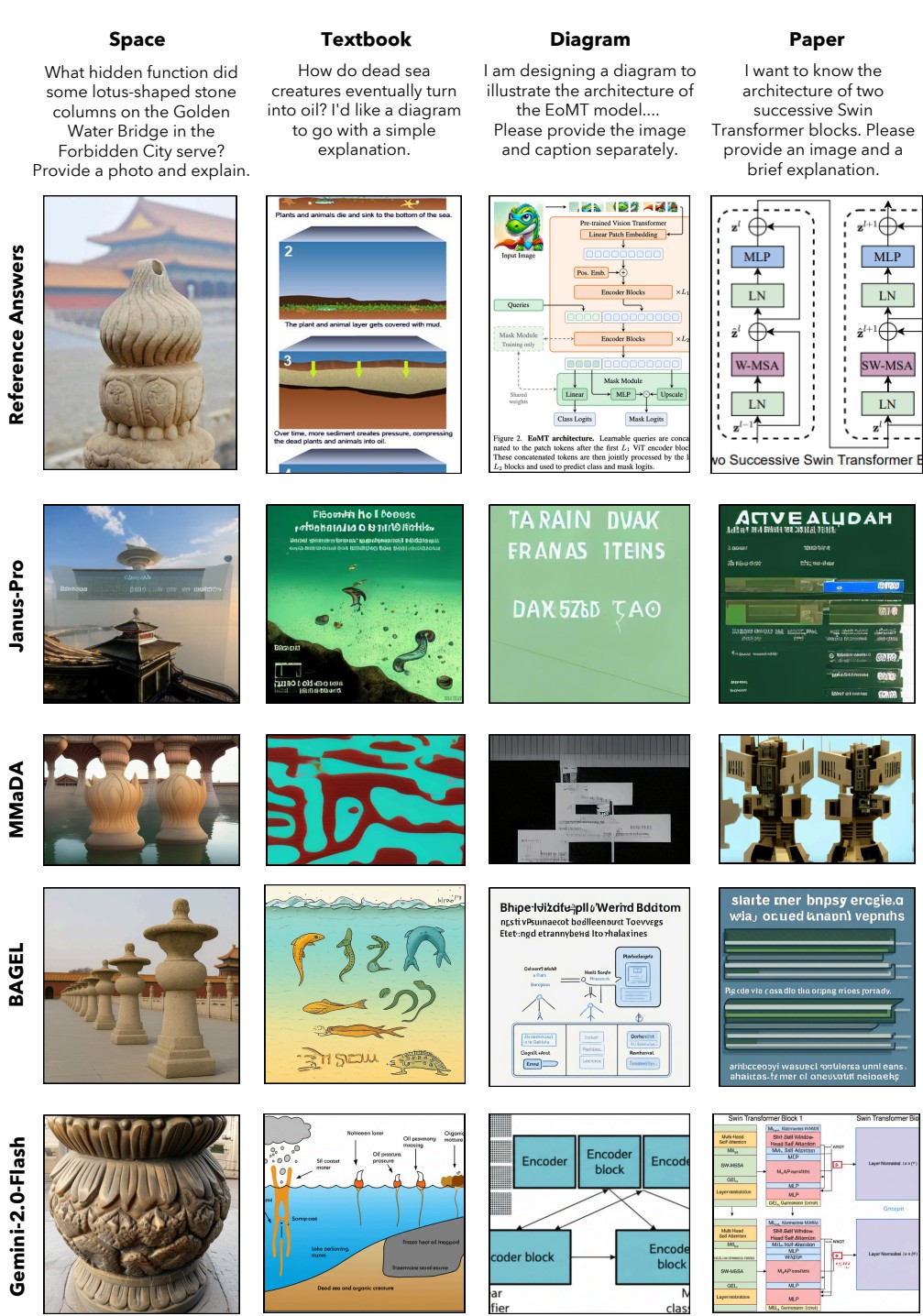

Figure 10: **Generated images on UEVAL's tasks with reference answers**. We visualize images generated by Janus-Pro, MMaDA, BAGEL, Gemini-2.0-Flash

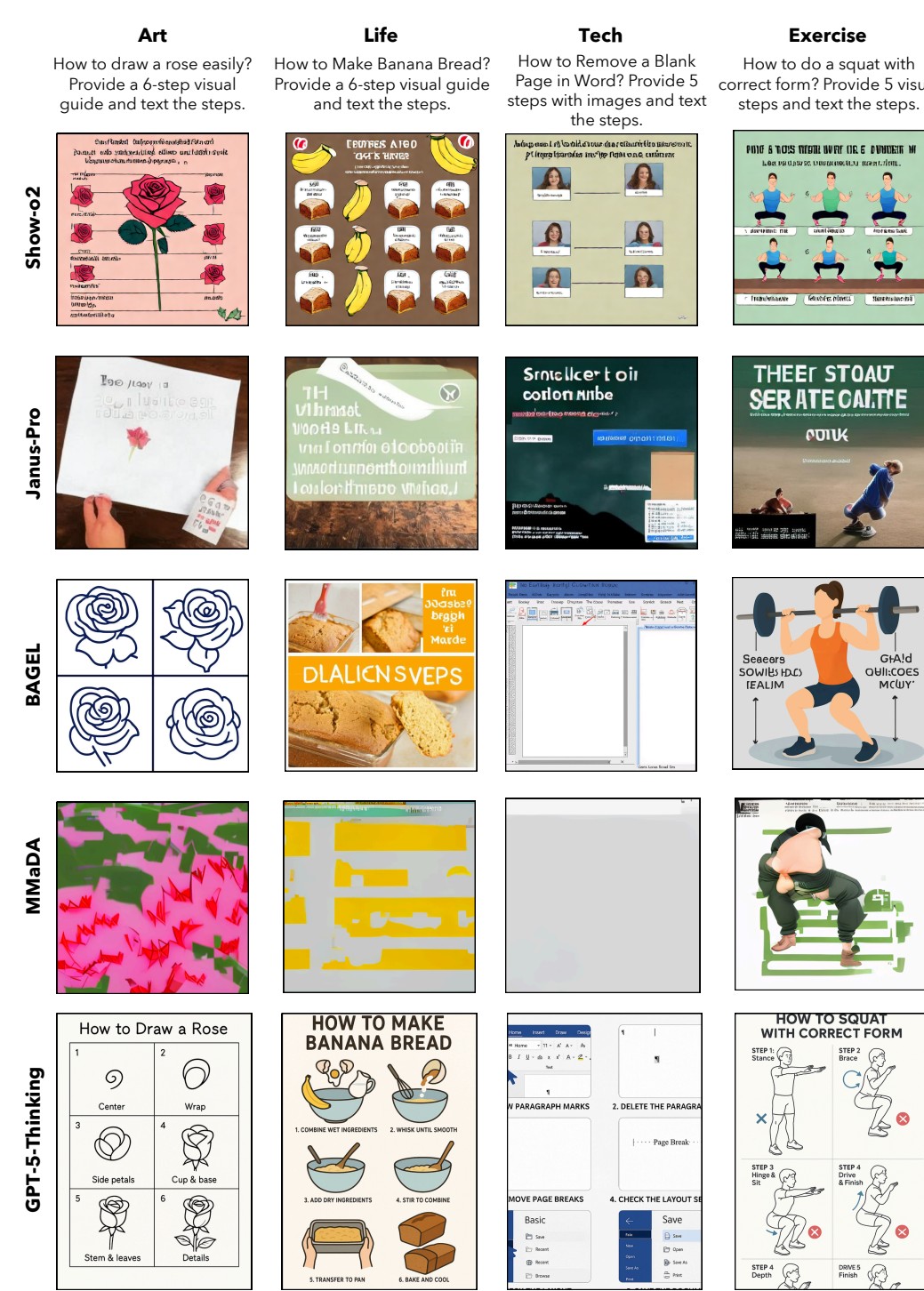

Figure 11: **Generated images on UEVAL's tasks without reference answers**. We prompt Show-o2, Janus-Pro, BAGEL, MMaDA, GPT-5-Thinking to produce step-by-step visual guides for each task.

# E  THE USE OF LARGE LANGUAGE MODELS (LLMS)

LLMs were only used to help polish the writing in this paper by suggesting better word choices at a sentence- or paragraph-level. We did not use LLMs to do anything beyond that, including drafting any paragraph in the paper, retrieving related work, or generating novel research idea.

