# OpenReview forum: "UEval: A Real-World Benchmark for Unified Multimodal Generation"
_ICLR.cc/2026/Conference — Submitted to ICLR 2026_

### Official Review · Reviewer_PWeU · 2025-10-16

**Soundness:** 2
**Presentation:** 3
**Contribution:** 2
**Rating:** 4
**Confidence:** 4

**Summary:**

This paper introduces UEVAL, a new benchmark for evaluating unified multimodal models (i.e., models that can generate both images and text simultaneously). The benchmark consists of 1,000 expert-curated prompts from 8 real-world domains. Its core contribution is the introduction of an innovative, data-dependent rubric-based evaluation framework to replace traditional methods. Through an evaluation of 9 leading models, the paper reveals significant challenges posed by this task to current models and highlights the critical role of "reasoning" in complex multimodal generation.

**Strengths:**

1.The paper accurately identifies and addresses a significant gap in the current multimodal evaluation landscape: the lack of effective metrics for unified image-text generation. The proposed rubric-based evaluation method, which customizes scoring criteria for each prompt, represents a novel and important directional attempt to move beyond the paradigm of a simple "LLM-as-a-judge".

2.The paper conducts a broad evaluation of current mainstream models, covering 9 representative models from both open-source and proprietary domains . Its results clearly reveal the performance bottlenecks of existing unified multimodal models (especially open-source ones) on this task and experimentally quantify the effectiveness of "reasoning traces". These findings themselves are of significant reference value.

**Weaknesses:**

1.Questionable Task Formulation:

The formulation of some prompts is mismatched with the visual medium required for the answer. For "Why" questions that require explaining internal structures or causality (e.g., the Statue of Liberty example), a single static image is inherently insufficient to provide a complete explanation. The paper's own analysis using a "reasoning trace" (Figure 5) inadvertently demonstrates that a truly sufficient answer requires a more complex format, such as a multi-step visual narrative (i.e., multiple images) accompanied by detailed explanations, rather than a single static image. This exposes an internal contradiction in some of the task designs.

2.Lack of Methodological Transparency:

The process for rubric creation is opaque. The paper states that rubrics are refined by "human experts" but provides no manual or guidelines that directed them, nor does it specify the experts' qualifications. This compromises the credibility and reproducibility of the scoring criteria.

Information on the human validation is insufficient. The validation was performed on only a 10% random sample of the data, which may be too small to draw robust conclusions for a 1,000-item benchmark. Furthermore, the professional backgrounds of the "human annotators" are not disclosed. The scatter plot in Figure 8, despite a moderate correlation ($\rho=0.76$), shows significant variance in individual cases, which, combined with the small sample size, weakens the claim of the automated framework's reliability.

3.Limitations of the Evaluation Framework:

"One-Size-Fits-All" Evaluation: The paper applies a uniform LLM-judge framework across 8 vastly different domains, from "Art" tutorials to "Paper" diagrams15. For highly structured and logical domains like "Diagram," this generic approach may fail to capture specific evaluation dimensions and does not incorporate domain-specific, rule-based metrics.

Evaluation Rigidity: For "guide" tasks that lack reference answers, the rigid requirement of a fixed number of steps may unfairly penalize creative or more effective answers that use a different structure.

Missing Analysis Dimensions: The results analysis focuses heavily on image generation errors  but lacks a concrete analysis of text-based errors and fails to adequately discuss the evaluation results for the core metric of image-text consistency.

**Questions:**

1.Regarding Task Design: How did you ensure that the benchmark's prompts, particularly "Why" questions, are fundamentally answerable with a single image-text pair? Was there a screening criterion to determine the suitability of this format for tasks requiring process or internal structural explanations?

2.Regarding Methodological Transparency: Could you provide or elaborate on the manual/guidelines given to the "human experts" for refining the rubrics? Furthermore, what professional qualifications did these experts and the "human annotators" for the 10% validation possess, especially for specialized domains like "Paper" and "Diagram"?

3.Regarding the Evaluation Framework: Considering the significant differences between domains like "Art" and "Diagram," do you believe a single LLM-based framework is sufficient for fair and accurate evaluation across all tasks? Have you considered incorporating rule-based, domain-specific metrics for tasks with objective correctness criteria? Additionally, for the "guide" tasks, how does the scoring system account for high-quality responses that deviate from the prompted number of steps?

4.Regarding the Depth of Analysis: The analysis focuses on image-based failures. Could you provide some typical examples of text-based errors made by the models? Furthermore, how did the models perform specifically on the critical metric of image-text consistency, and can you share any quantitative analysis or illustrative case studies on this?

---

> ### Author Response · Authors · 2025-11-27
>
> We appreciate that the reviewer recognizes rubric-based evaluation as an important missing component in unified image–text generation, and we are encouraged that our benchmark helps fill this gap by providing effective metrics. We address your questions and concerns below:
>
> > W1:The formulation of some prompts is mismatched with the visual medium required for the answer. For "Why" questions that require explaining internal structures or causality (e.g., the Statue of Liberty example), a single static image is inherently insufficient to provide a complete explanation. The paper's own analysis using a "reasoning trace" (Figure 5) inadvertently demonstrates that a truly sufficient answer requires a more complex format, such as a multi-step visual narrative (i.e., multiple images) accompanied by detailed explanations, rather than a single static image. This exposes an internal contradiction in some of the task designs.
>
> Thank you for raising this point. We agree that a single static image may not always provide a fully comprehensive explanation. However, we would like to clarify that under our rubric-based evaluation, the curated reference image–text answers achieve **94.5** average scores averaged across all tasks. This indicates that, in practice, **generating a single well-chosen image is sufficient to address the question for the types of tasks included in our benchmark.**
>
> In addition, requiring multiple images in the reference answers would significantly increase both the complexity and the ambiguity of constructing high-quality ground-truth references. Since we rely on LLM-as-a-judge to automate scoring, introducing multi-image inputs would introduce further challenges: **current multimodal models exhibit weaker multi-image reasoning compared to single-image understanding([1][2][3])**. For these reasons, we intentionally adopt the single-image setup to maintain clarity, consistency, and reliability in evaluation.
>
> [1] Wang, Fei, et al. "MuirBench: A Comprehensive Benchmark for Robust Multi-image Understanding."ICLR 2025.
>
> [2] Fu, Xingyu, et al. "Blink: Multimodal large language models can see but not perceive." ECCV 2024.
>
> [3] Jiang, Dongfu, et al. "Mantis: Interleaved Multi-Image Instruction Tuning." Transactions on Machine Learning Research.
>
> > W2.1: The process for rubric creation is opaque. The paper states that rubrics are refined by "human experts" but provides no manual or guidelines that directed them, nor does it specify the experts' qualifications. This compromises the credibility and reproducibility of the scoring criteria.
>
> Thank you for raising this point. We agree that more details on our human annotation setup will strengthen our paper, and we have included these details in Section 2.2 of our current draft.
>
> **For each sample in our benchmark, a primary annotator drafts the question, reference answer, and initial rubric, after which three co-authors independently verify the annotation.** All annotators—including the primary annotator and the verifying co-authors—have substantial experience in building multimodal benchmarks. Human annotators control the benchmark construction at a system level, including the input (question prompt), the output (reference answers), and the grading criteria (rubrics). Our goal is to have rubrics assigning high scores to the reference answers while appropriately penalizing incorrect or misaligned model outputs. Only rubric items that all reviewers agree are clear, measurable, and aligned with the task design are retained in the final benchmark.
>
>  We have incorporated the above discussion as a new paragraph in Section 2.2 (Human review).
>
> >W2.2: Information on the human validation is insufficient. The validation was performed on only a 10% random sample of the data, which may be too small to draw robust conclusions for a 1,000-item benchmark
>
> Thank you for pointing this out. We agree that a single 10% random sample may not be sufficient to demonstrate the robustness of human validation. To address this concern, we perform two additional independent 10% random samples, resulting in three non–overlapping validation rounds.
>
> The results are highly consistent:
>
> | Validation Round | Agreement (%) | Pearson Correlation (ρ) |
> |------------------|----------------------------|---------------------------|
> | Round 1          | 89.7%                      | 0.81                      |
> | Round 2          | 90.4%                      | 0.84                      |
> | Round 3          | 91.2%                      | 0.88                      |

---

> > ### Author Response · Authors · 2025-11-27
> >
> > > W2.3: Furthermore, the professional backgrounds of the "human annotators" are not disclosed.
> >
> > Thank you for pointing this out. We agree that clarifying the professional backgrounds of the human annotators improves transparency. All human annotators involved in the rubric verification and the 10% human evaluation are the four co-authors of the paper, each with substantial experience in multimodal evaluation, dataset annotation, and LLM benchmarking.
> >
> > >W2.4:The scatter plot in Figure 8, despite a moderate correlation, shows significant variance in individual cases, which, combined with the small sample size, weakens the claim of the automated framework's reliability.
> >
> > Thank you for the comment. In response to feedback from multiple reviewers, **we have revised several tasks’ questions and their corresponding rubrics.** These revisions include removing numerical constraints (e.g., fixed step numbers) and adding reference images and reference text for open-ended tasks. We regenerated rubrics accordingly and re-ran the human evaluation, with the updated results reported in W2.2.
> >
> > >W3.1:"One-Size-Fits-All" Evaluation: The paper applies a uniform LLM-judge framework across 8 vastly different domains, from "Art" tutorials to "Paper" diagrams15. For highly structured and logical domains like "Diagram," this generic approach may fail to capture specific evaluation dimensions and does not incorporate domain-specific, rule-based metrics.
> >
> > Thank you for raising this point. **In UEval,  these domains are not evaluated purely through a generic, one-size-fits-all judge.** Instead, this task includes gold answers—both the reference image and the reference text—which provide concrete ground-truth signals for generating precise, task-specific rubrics.
> >
> > Because the rubrics are conditioned on the gold answer, **they naturally incorporate the domain-specific constraints required for structured tasks.** For example, in Diagram tasks, the rubric we generate includes highly specialized requirements such as:
> >
> > “The image must include all specified core components: Input Image, Pretrained Teacher, Student Model, multiple Rate-Distortion Modules (RDMs).”
> >
> > “The image must accurately depict the data flow: an input image is fed to the teacher, which generates embeddings for the RDMs; the RDMs' outputs then supervise the student model.”
> >
> > These constraints are not generic—they reflect exactly the structure required in technical diagram generation tasks, and they arise from conditioning the rubric on task-specific gold answers. Therefore, while UEVAL uses a unified framework, the rubrics themselves are domain-specialized.
> >
> > > W3.2:For "guide" tasks that lack reference answers, the rigid requirement of a fixed number of steps may unfairly penalize creative or more effective answers that use a different structure.
> >
> > Thank you for pointing this out — we agree with the concern. For guide tasks, enforcing a fixed number of steps could unfairly penalize creative or more effective responses that use different structures.
> >
> > **To address this, we have revised both the question design and the corresponding rubrics for guide tasks.** The updated version no longer requires a fixed number of steps, and we have re-evaluated all models under the updated questions and rubrics to ensure fairness and consistency across the benchmark.
> >
> > **In addition, we significantly improved the rubric design by introducing reference images and reference text for open-ended guide tasks.** These references provide annotators with a clear and consistent anchor for interpreting the intended process, preventing the rubric from drifting toward subjective stylistic expectations.
> >
> > **These changes substantially increase the clarity and objectivity of the evaluation.** For example, in the original rubric, criteria such as “The subject of the drawing must be a ‘cartoon cat’, and the visual style must be consistent across all five images” introduced significant ambiguity, because “style consistency” has no precise operational definition. The automatic judge often enforced this requirement rigidly—flagging outputs as inconsistent due to minor differences in facial features—whereas human annotators typically judged the overall style to be coherent, leading to substantial disagreement.
> >
> > In the updated rubric, we removed such subjective style-based requirements and replaced them with fully objective, structure-oriented criteria, such as:
> > * “Each image must correspond exactly to its associated textual step.”
> >
> > * “Newly added lines or shapes must be distinguishable from the previous step.”
> >
> > * “The image sequence must illustrate a coherent progression from basic shapes to the final drawing.”
> >
> > By grounding the rubric in reference answers, the revised evaluation eliminates subjective ambiguity and leads to much higher human–rubric alignment.

---

> > > ### Author Response · Authors · 2025-11-27
> > >
> > > >W3.3:The results analysis focuses heavily on image generation errors but lacks a concrete analysis of text-based errors and fails to adequately discuss the evaluation results for the core metric of image-text consistency.
> > >
> > > Thank you for the comment. In addition to image-generation issues, our evaluation also reveals several systematic categories of text-based errors, which play a central role in multimodal reasoning quality.
> > >
> > >
> > > For example, in the Art task, **Bagel produces step-by-step instructions that are logically inconsistent.** The model asks the user to “draw the eyes” in Step 1 but unexpectedly repeats the same instruction in Step 4, and mentions adding whiskers in Steps 1, 2, 5, and 6.  In the Space task, the model often fails to provide the causal explanations required by the task. For the Statue of Liberty question, the model describes historical and symbolic information but omits core mechanisms explicitly required by the rubric, such as the internal spiral staircase.
> > >
> > >
> > > **Regarding image–text consistency, this metric is indeed one of the core evaluation dimensions in UEval**, and it is explicitly encoded in the rubrics through several concrete criteria—for example, whether each image aligns with its corresponding textual step and whether newly added visual elements match the textual description. **Our evaluation also reveals typical failure cases on this dimension.** For instance, in the Art task (e.g., the cartoon cat drawing), Bagel frequently produces images that do not match the textual instructions—such as adding whiskers or facial details that the text never mentions, or omitting visual elements that the text explicitly requires.
> > >
> > > >Q1:Regarding Task Design: How did you ensure that the benchmark's prompts, particularly "Why" questions, are fundamentally answerable with a single image-text pair? Was there a screening criterion to determine the suitability of this format for tasks requiring process or internal structural explanations?
> > >
> > > Please see our response to W1.
> > >
> > > >Q2:Regarding Methodological Transparency: Could you provide or elaborate on the manual/guidelines given to the "human experts" for refining the rubrics? Furthermore, what professional qualifications did these experts and the "human annotators" for the 10% validation possess, especially for specialized domains like "Paper" and "Diagram"?
> > >
> > > Please see our response to W2.1.
> > >
> > >
> > > >Q3:Regarding the Evaluation Framework: Considering the significant differences between domains like "Art" and "Diagram," do you believe a single LLM-based framework is sufficient for fair and accurate evaluation across all tasks? Have you considered incorporating rule-based, domain-specific metrics for tasks with objective correctness criteria? Additionally, for the "guide" tasks, how does the scoring system account for high-quality responses that deviate from the prompted number of steps?
> > >
> > > Please see our response to W3.1.
> > >
> > >
> > > >Q4.1:Regarding the Depth of Analysis: The analysis focuses on image-based failures. Could you provide some typical examples of text-based errors made by the models?
> > >
> > > Please see our response to W3.3.
> > >
> > >
> > > >Q4.2:Furthermore, how did the models perform specifically on the critical metric of image-text consistency, and can you share any quantitative analysis or illustrative case studies on this?
> > >
> > > Please see our response to W3.3.

---

> > > > ### Comment · Reviewer_PWeU · 2025-11-27
> > > >
> > > > Thank you for your detailed reply! The substantial examples have resolved most of my questions, and I will adjust my score upward.

---

### Official Review · Reviewer_iSKG · 2025-10-29

**Soundness:** 3
**Presentation:** 2
**Contribution:** 3
**Rating:** 4
**Confidence:** 3

**Summary:**

This paper introduces UEVAL, a challenging benchmark for evaluating unified multimodal generation models that produce both images and text in response to complex queries. Comprising 1,000 expert-curated prompts from 8 real-world domains (e.g., space, textbook, diagram, art), UEVAL requires models to generate interleaved multimodal outputs, addressing gaps in existing evaluations focused on VQA or text-to-image tasks. The authors propose a rubric-based scoring system where LLMs generate initial rubrics based on reference answers, refined by humans for reliability, resulting in 8.1K criteria. Experiments on 9 models reveal high difficulty, with GPT-5-Thinking scoring 66.6/100 and the best open-source model at 22.4/100, highlighting the benefits of reasoning traces in improving generation quality. Contributions include the benchmark dataset, rubric framework, and insights into reasoning's role in multimodal tasks.

**Strengths:**

The paper demonstrates strong originality by formulating a novel benchmark for unified multimodal generation, creatively combining real-world scenarios with data-dependent rubrics to evaluate interleaved image-text outputs, which extends beyond traditional VQA or T2I paradigms and addresses a clear gap in assessing complex multimodal reasoning. In terms of quality, the benchmark's construction is rigorous, with 1,000 diverse prompts sourced from expert curation and validated references, supported by comprehensive experiments on both proprietary and open-source models that provide actionable insights, such as the impact of reasoning traces on non-reasoning models. Clarity is a highlight, as the paper is well-structured with intuitive figures (e.g., Figure 1 illustrating task distributions) and detailed appendices, making the rubric generation and evaluation pipelines easy to follow. Finally, the significance is evident in its potential to drive advancements in unified models, offering a scalable, reproducible evaluation framework that emphasizes reasoning's importance, with public release of data and code enhancing community impact.

**Weaknesses:**

One key weakness is the benchmark's limited scale and diversity; with only 1,000 prompts across 8 domains, it may not fully capture the breadth of real-world multimodal tasks, such as those involving dynamic video inputs or non-English languages—to improve, the authors could expand the dataset by incorporating user-generated prompts from broader sources like social media datasets and validate cross-cultural applicability. Another issue is the heavy reliance on proprietary LLMs (e.g., Gemini-2.5-Pro) for rubric generation and judging, which introduces potential biases and reproducibility challenges; a constructive step would be to include ablation studies using open-source alternatives and quantify inter-model agreement metrics beyond the reported Pearson correlation. Additionally, the analysis of reasoning benefits is preliminary and lacks deeper mechanistic insights, such as why open-source models like BAGEL fail to improve with transferred traces—future work could address this by conducting fine-grained error analyses or probing model internals to identify specific failure modes in multimodal token prediction.

**Questions:**

1. The paper emphasizes reasoning traces' benefits for proprietary models but not open-source ones—could you elaborate on potential architectural differences (e.g., tokenization strategies or multimodal fusion mechanisms in BAGEL vs. GPT-5) that might explain this disparity, and provide any additional experiments or hypotheses that could clarify if this is due to training data quality or model capacity? A detailed response here could strengthen the claim about reasoning's universality in unified models.

2. Your rubric-based evaluation shows strong human-LLM alignment (Pearson ρ=0.76), but how robust is this to variations in the judge model? For instance, if you replaced Gemini-2.5-Pro with an open-source alternative like LLaVA, would the scores change significantly, and what metrics (e.g., Cohen's kappa) did you compute to assess this? Addressing this could alleviate concerns about evaluation bias and enhance the framework's generalizability.

3. UEVAL excludes input images in prompts, focusing solely on text-based queries—how would incorporating visual inputs (e.g., for tasks like diagram completion or editing) affect model performance, and do you have preliminary data or plans to extend the benchmark in this direction? This could address a noted limitation and potentially reveal new insights into multimodal reasoning chains.

---

> ### Author Response · Authors · 2025-11-27
>
> We are glad that the reviewer finds our benchmark has strong originality by formulating a novel benchmark for unified multimodal generation and extending beyond traditional VQA or T2I paradigms to address a clear gap in assessing complex multimodal reasoning. We address your questions and concerns below:
>
> >W1:One key weakness is the benchmark's limited scale and diversity; with only 1,000 prompts across 8 domains, it may not fully capture the breadth of real-world multimodal tasks, such as those involving dynamic video inputs or non-English languages—to improve, the authors could expand the dataset by incorporating user-generated prompts from broader sources like social media datasets and validate cross-cultural applicability.
>
> We agree that the scale and diversity of the current benchmark could be expanded. However, we would like to emphasize that annotating the entire dataset is a time-consuming process that requires significant human effort. Given the limited resources of our team, we have focused on creating a high-quality, representative sample of tasks to maintain the reliability and validity of the benchmark.
>
> We would also like to emphasize that, despite its moderate size, our benchmark fills an important gap in the existing landscape: **it is the first benchmark specifically designed to evaluate complex multimodal reasoning, and it provides over 11,000 high-quality rubrics, enabling a level of fine-grained, procedural evaluation that has not been available in prior multimodal datasets.**
> We fully acknowledge the value of expanding the dataset to cover additional domains and we plan to explore these directions in future iterations. Nevertheless, we believe the current benchmark already offers a unique and valuable platform.

---

> > ### Author Response · Authors · 2025-11-27
> >
> > >W2:Another issue is the heavy reliance on proprietary LLMs (e.g., Gemini-2.5-Pro) for rubric generation and judging, which introduces potential biases and reproducibility challenges; a constructive step would be to include ablation studies using open-source alternatives and quantify inter-model agreement metrics beyond the reported Pearson correlation.
> >
> > We consider the generated rubrics as part of our benchmark, and **we will release the complete set of finalized, human-verified rubrics in our final datasets.** For this reason, we do not plan to add additional experiments comparing different rubric-generation models in the paper. That said, we fully agree that understanding how different models influence rubric construction and scoring is an important direction. We hope future work will explore this systematically—for example, by evaluating how rubric-generation models affect the reliability and fairness of automated assessment.
> >
> > Below, we also compare different models used as LLM-as-a-judge. In addition, we conducted experiments with Qwen3-VL-8B, InternVL-3.5-9B, and GLM-4.1-9B-Thinking, Qwen3-vl-235B, Doubao-seed-1.6vision, GPT-5 to assess their rubric-judging performance.
> > | Judge Model            | Space Image | Space Text | Textbook Image | Textbook Text | Diagram Image | Diagram Text | Paper Image | Paper text |
> > |------------------------|-------------|------------|-----------------|----------------|----------------|---------------|----------------|---------------|
> > | Gemini-2.5-pro         | 81.6       | 85.5      | 62.1          | 90.2          | 44.9          | 79.3         | 43.3 | 62.7|
> > | Qwen3-vl-8B            | 79.9      | 84.5     | 79.7      | 90.9          | 75.6          | 75.3         | 57.6 | 53.3 |
> > | Internvl3.5-8B         | 82.4       | 77.4      | 89.6          | 85.7          | 73.0          | 63.4        | 60.3 | 46.9|
> > | GLM-4.1-thinking       | 84.3       | 84.3      | 79.4          | 87.9          | 67.1          | 69.2         | 50.0 | 49.6|
> > | Qwen3-vl-235B           | 81.5       | 82.6      | 80.7          | 89.8          | 69.7          | 74.9         | 52.4 | 54.8|
> > | Doubao-seed-1.6vision  | 86.9       | 84.1      | 71.6          | 90.4          | 67.1          | 69.2         | 51.6 | 56.0|
> > | GPT-5    | 76.9       | 83.0      |        56.4      |    89.5         |  35.3        |   76.4        |  37.6  | 55.0 |
> >
> >
> > | Judge Model            | Life Image | Life Text | Exercise Image | Exercise Text | Art Image | Art Text |Tech Image | Tech Text |
> > |------------------------|------------|-----------|----------------|----------------|-----------|----------|-----------|----------|
> > | Gemini-2.5-pro         | 30.7      | 78.4     | 33.1          | 78.0         | 23.8      | 91.9    | 18.9 | 83.3 |
> > | Qwen3-vl-8b            | 77.5      | 81.1     | 76.4          | 77.2         | 78.0      | 90.9    | 75.0 | 81.5 |
> > | Internvl3.5-8b         | 68.3      | 76.6     | 75.7          | 69.4          | 51.1     | 86.5    |67.9 | 74.9|
> > | GLM-4.1-thinking       | 70.8     | 78.2     | 68.9          | 72.5         | 71.5     | 87.2    |69.9 | 78.7 |
> > | Qwen3-vl-235B           | 42.9      | 79.8     | 49.9          | 76.2         | 58.0     | 89.4    | 34.3 | 81.1 |
> > | Doubao-seed-1.6vision  | 61.6      | 79.9     | 65.4          | 76.4          | 59.9     | 90.4    |55.0 | 80.7|
> > | GPT-5   |       23.6  |    76.6   |  32.2    |  71.2           | 25.5     | 87.4    | 11.3     | 79.5    | 11.3 | 79.5|
> >
> >
> > We observe that the strongest judges—GPT-5 and Gemini-2.5-Pro—produce highly consistent scores across all tasks. In contrast, current open-weights multimodal models (e.g. Qwen3-VL-8B) remain considerably limited as judges.
> >
> > **To further ensure reproducibility, we will release all model inference outputs.** This allows future, more capable judge models to re-evaluate the exact same set of outputs without requiring any regeneration.

---

> > > ### Author Response · Authors · 2025-11-27
> > >
> > > >W3:Additionally, the analysis of reasoning benefits is preliminary and lacks deeper mechanistic insights, such as why open-source models like BAGEL fail to improve with transferred traces—future work could address this by conducting fine-grained error analyses or probing model internals to identify specific failure modes in multimodal token prediction.
> > >
> > > Thank you for the comment. We conduct a detailed error analysis on BAGEL’s outputs. We randomly sample 10% of the data and manually annotate each item according to 7 error patterns. The overall error distribution is shown in Figure (https://drive.google.com/file/d/1YsNGNcZPUOO9P1od0IKkk3WF8L6gVxlV/view?pli=1)
> > >
> > > **From the distribution, it is clear that BAGEL’s image generation capability is still limited.** Most failures come from missing key entities, incorrect content selection, image–text misalignment, and low-level visual artifacts. These issues indicate that BAGEL often cannot generate the structural visual elements required to answer the question. Since these failures occur at the visual token generation stage, the additional reasoning trace cannot meaningfully influence or correct the final image output.
> > >
> > > We additionally provide qualitative examples of both image-generation and text-generation failures in Figure (https://drive.google.com/file/d/1r6rxmGEStVSRE692591sWOaSg0hPZKgr/view?usp=sharing)
> > >
> > > >Q1:The paper emphasizes reasoning traces' benefits for proprietary models but not open-source ones—could you elaborate on potential architectural differences (e.g., tokenization strategies or multimodal fusion mechanisms in BAGEL vs. GPT-5) that might explain this disparity, and provide any additional experiments or hypotheses that could clarify if this is due to training data quality or model capacity? A detailed response here could strengthen the claim about reasoning's universality in unified models.
> > >
> > > Thank you for the thoughtful comment. We have added additional analysis to clarify why reasoning traces noticeably benefit proprietary unified models (GPT-5-Instant, Gemini-2.5-Flash) but not the open-source unified model BAGEL.
> > >
> > > **First, we believe training data differences are a major factor.** Proprietary models might be trained on large amounts of interleaved image–text data. This enables the model to learn how to adjust image generation based on detailed reasoning. In contrast, BAGEL relies heavily on caption/alt-text–style data, which are short, weakly aligned, and rarely contain reasoning chains. As a result, when presented with a long reasoning trace, BAGEL often cannot utilize the information and may even treat it as an unusual or noisy conditioning signal.
> > >
> > > **Second,  BAGEL’s visual generation remains weak (frequent missing structures, artifacts).** Combined with its smaller parameter count and multimodal context budget compared to proprietary models, essential visual information is often lost before the reasoning trace could meaningfully influence image tokens. **Thus, the reasoning trace cannot deliver the same benefits for BAGEL as it does for larger, better-supervised proprietary models.**
> > >
> > > >Q2:Your rubric-based evaluation shows strong human-LLM alignment (Pearson ρ=0.76), but how robust is this to variations in the judge model? For instance, if you replaced Gemini-2.5-Pro with an open-source alternative like LLaVA, would the scores change significantly, and what metrics (e.g., Cohen's kappa) did you compute to assess this? Addressing this could alleviate concerns about evaluation bias and enhance the framework's generalizability.
> > >
> > > Please see our response to W2.
> > >
> > > >Q3:UEVAL excludes input images in prompts, focusing solely on text-based queries—how would incorporating visual inputs (e.g., for tasks like diagram completion or editing) affect model performance, and do you have preliminary data or plans to extend the benchmark in this direction? This could address a noted limitation and potentially reveal new insights into multimodal reasoning chains.
> > >
> > > Thank you for highlighting this point. We agree that excluding input images from prompts is a limitation of the current version of UEVAL. **Our goal in this release is to first establish a high-quality benchmark to evaluate multimodal generation and challenge current models to reasoning in both image and texts.** In future iterations, we plan to expand UEVAL to include tasks that incorporate input images and cover a broader range of multimodal reasoning capabilities.

---

### Official Review · Reviewer_eXoG · 2025-10-29

**Soundness:** 4
**Presentation:** 4
**Contribution:** 3
**Rating:** 8
**Confidence:** 4

**Summary:**

The paper introduces a new benchmark for evaluating multimodal generation. Per-sample curated rubric based LLM-as-judge evaluations are proposed as an alternative to traditional single-prompt LLM-as-judge techniques. Proprietary models reach 66% accuracy, while open source models perform worse, highlighting the benchmark's difficulty.

**Strengths:**

1.	The proposed idea of generating a per-sample rubric for evaluating multimodal generation is novel and tries to solve the instability issues observed with standard single-prompt multimodal LLM-as-judge evaluations.
2.	Both open-source and proprietary models score modestly on the benchmark, proving that this is a challenging benchmark for current state-of-the-art models in multimodal generation.
3.	The methodology of generating rubrics with a strong multimodal LLM followed by human verification is sound.
4.	Qualitative analysis and discussions of model behavior patterns is well presented. Alignment of proposed metric with human judgements along with alignment of rubric scoring with reference images are present, making the work well rounded.

**Weaknesses:**

1.	The paper states that pairwise win-rate judging is unstable, and a single prompt LLM-as-judge evaluation overlooks sample specific differences. Some examples/studies demonstrating these deficiencies of current standard methods can improve the motivation.
2.	Gemini-2.5-Pro, a proprietary model is used as the judge. This hurts reproducibility since API based models can change over time. It would be interesting to see results using leading open-weights multimodal models as the judge.
3.	In open-ended tasks like “art” where there could be multiple correct answers, a discussion of whether the rubrics along with the judge used lead to fair evaluations is missing. Deeper analysis of the low human judgement alignment scores for such tasks would also be useful.

**Questions:**

1.	Can leading open-weights multimodal models replace Gemini-2.5-Pro as the judge? Do they show similar high correlation with human judgement? If not, what are the shortcomings of using them?
2.	Could the authors provide a deeper analysis of how the judgements for “art” varies between LLMs and humans including fairness and biases of judge models in such open ended cases?

---

> ### Author Response · Authors · 2025-11-27
>
> We are happy that the reviewer finds our benchmark as a challenging benchmark for current state-of-the-art models in multimodal generation. We address your questions and concerns below:
>
> > W1:The paper states that pairwise win-rate judging is unstable, and a single prompt LLM-as-judge evaluation overlooks sample specific differences. Some examples/studies demonstrating these deficiencies of current standard methods can improve the motivation.
>
> Thank you for raising this important point. **Pairwise win-rate judging often requires hundreds of pairwise samples**, which makes it very difficult for model developers to execute in practice. ChatbotArena [1] and OpenING [2] utilize this pairwise evaluation method to assess models.
>
> For single prompt evaluation, we select a random 10% subset of the data for comparison. This subset is evaluated using gemini-2.5-pro under the single prompt LLM-as-judge evaluation setting, and resulting scores are compared with human evaluations. The correlation is weak: **we computed a Pearson correlation of r = 0.3922, indicating that the single-prompt method fails to track human judgment reliably**.
>
> We observed a severe lack of score dispersion. For space task, the single prompt evaluation assigns near-perfect scores to almost all model outputs.
>
> [1] Chatbot Arena  https://lmarena.ai/
>
> [2] Zhou, Pengfei, et al. "OpenING: A Comprehensive Benchmark for Judging Open-ended Interleaved Image-Text Generation." CVPR. 2025.
>
> >W2:Gemini-2.5-Pro, a proprietary model is used as the judge. This hurts reproducibility since API based models can change over time. It would be interesting to see results using leading open-weights multimodal models as the judge.
>
> Thank you for raising this concern. We agree that relying solely on a proprietary model may affect reproducibility, as API-based models can evolve over time. To address this, **we have added leading open-weights multimodal models as additional judges in our evaluation**. In our experiments, we used the following open-weights models and frontier models: Qwen3-VL-8B, InternVL-3.5, GLM-4.1V-9B-Thinking, Qwen3-VL-235B, Doubao-Seed-1.6Vision, and GPT-5 on GPT5-thinking results. Here are the results:
>
> | Judge Model            | Space Image | Space Text | Textbook Image | Textbook Text | Diagram Image | Diagram Text | Paper Image | Paper text |
> |------------------------|-------------|------------|-----------------|----------------|----------------|---------------|----------------|---------------|
> | Gemini-2.5-pro         | 81.6       | 85.5      | 62.1          | 90.2          | 44.9          | 79.3         | 43.3 | 62.7|
> | Qwen3-vl-8b            | 79.9      | 84.5     | 79.7      | 90.9          | 75.6          | 75.3         | 57.6 | 53.3 |
> | Internvl3.5-8b         | 82.4       | 77.4      | 89.6          | 85.7          | 73.0          | 63.4        | 60.3 | 46.9|
> | GLM-4.1-thinking       | 84.3       | 84.3      | 79.4          | 87.9          | 67.1          | 69.2         | 50.0 | 49.6|
> | Qwen3-vl-235B           | 81.5       | 82.6      | 80.7          | 89.8          | 69.7          | 74.9         | 52.4 | 54.8|
> | Doubao-seed-1.6vision  | 86.9       | 84.1      | 71.6          | 90.4          | 67.1          | 69.2         | 51.6 | 56.0|
> | GPT-5    | 76.9       | 83.0      |        56.4      |    89.5         |  35.3        |   76.4        |  37.6  | 55.0 |
>
> | Judge Model            | Life Image | Life Text | Exercise Image | Exercise Text | Art Image | Art Text |Tech Image | Tech Text |
> |------------------------|------------|-----------|----------------|----------------|-----------|----------|-----------|----------|
> | Gemini-2.5-pro         | 30.7      | 78.4     | 33.1          | 78.0         | 23.8      | 91.9    | 18.9 | 83.3 |
> | Qwen3-vl-8b            | 77.5      | 81.1     | 76.4          | 77.2         | 78.0      | 90.9    | 75.0 | 81.5 |
> | Internvl3.5-8b         | 68.3      | 76.6     | 75.7          | 69.4          | 51.1     | 86.5    |67.9 | 74.9|
> | GLM-4.1-thinking       | 70.8     | 78.2     | 68.9          | 72.5         | 71.5     | 87.2    |69.9 | 78.7 |
> | Qwen3-vl-235B           | 42.9      | 79.8     | 49.9          | 76.2         | 58.0     | 89.4    | 34.3 | 81.1 |
> | Doubao-seed-1.6vision  | 61.6      | 79.9     | 65.4          | 76.4          | 59.9     | 90.4    |55.0 | 80.7|
> | GPT-5   |       23.6  |    76.6   |  32.2    |  71.2           | 25.5     | 87.4    | 11.3     | 79.5    | 11.3 | 79.5|
>
> **To further ensure reproducibility, we will release all model inference outputs**. This allows future, more capable judge models to re-evaluate the exact same set of outputs without requiring any regeneration.

---

> > ### Author Response · Authors · 2025-11-27
> >
> > >W3.1:In open-ended tasks like “art” where there could be multiple correct answers, a discussion of whether the rubrics along with the judge used lead to fair evaluations is missing.
> >
> > Thank you for raising this important point. We agree that open-ended tasks, such as the “art” task, where multiple correct answers are possible, it is crucial to ensure that the rubrics and the judging process lead to fair and consistent evaluations.
> >
> > In our submission, **we have designed the rubrics to focus on generalizable and task-agnostic evaluation dimensions**, such as step structure, text-image alignment, temporal coherence, rather than enforcing any single correct content.
> > For example, in the art task, for the question “How to draw a cat ?”, the rubric evaluates structural and procedural aspects of the output, such as:
> >
> > “The answer must describe a step-by-step process for drawing a cat.”,
> >
> > “The sequence of images must illustrate the complete drawing process from the initial basic shapes to the final version.”
> >
> > “Each image must directly correspond to a single, sequential step outlined in the text answer.”
> >
> > “Newly added lines or shapes in each step should be distinguishable from the previous steps.”
> >
> > These criteria do not prescribe what the cat must look like or which artistic style is “correct.” Instead, **they assess coherence, progression, and cross-modal alignment, ensuring that diverse but valid outputs are evaluated fairly.**
> >
> > > W3.2:Deeper analysis of the low human judgement alignment scores for such tasks would also be useful.
> >
> > Thank you for the suggestion. Following feedback from other reviewers, **we carefully revised the questions and rubrics for all tasks to improve clarity and ensure better alignment between human judgement and rubric-based scoring.** We improve the rubric design by removing numerical requirements (e.g., fixed step numbers) and introducing reference images and text,  which provide annotators with a clear and consistent anchor for interpreting the intended drawing process.
> >
> > **These changes substantially increased the clarity and objectivity of the evaluation.** For example, in the original rubric, criteria such as “The subject of the drawing must be a ‘cartoon cat’, and the visual style must be consistent across all five images” introduced significant ambiguity, as “style consistency” lacks a precise operational definition. The automatic judge often enforced this requirement rigidly—flagging outputs as inconsistent due to minor differences in facial features—whereas human annotators typically judged the overall style to be coherent, leading to substantial disagreement.
> >
> > In the updated rubric, we removed such subjective style-based requirements and replaced them with fully objective, structure-oriented criteria, including:“each image must correspond exactly to its associated textual step” ，“newly added lines or shapes must be distinguishable from the previous step” and “the image sequence must illustrate a coherent progression from basic shapes to the final drawing”. **Rubric based on reference answers eliminates subjective ambiguity and results in much higher human–rubric alignment.**

---

> > > ### Author Response · Authors · 2025-11-27
> > >
> > > >Q1:Can leading open-weights multimodal models replace Gemini-2.5-Pro as the judge? Do they show similar high correlation with human judgement? If not, what are the shortcomings of using them?
> > >
> > > Thank you for the question. To assess whether leading open-weights multimodal models can replace Gemini-2.5-Pro as judges, we evaluated Qwen3-VL-8B, InternVL-3.5, GLM-4.1V-9B-Thinking, Qwen3-VL-235B, Doubao-Seed-1.6Vision, and GPT-5 as alternative judges on generated images and text from GPT5-thinking. We additionally conducted human evaluation on a 10% subset of the benchmark and directly compared each model’s scores with human judgements.
> > >
> > > Here are the results:
> > > | Judge Model            | Agreement (%) | Correlation (ρ) |
> > > |------------------------|----------------|------------------|
> > > | Gemini-2.5-Pro         |          91.2      |          0.88        |
> > > | Qwen3-VL-8B            |      78.6      |           0.54       |
> > > | InternVL-3.5-8B        |           74.8     |        0.45          |
> > > | GLM-4.1V-9B-Thinking   |          68.1      |         0.36         |
> > > | Qwen3-VL-235B           |        80.3        |           0.57       |
> > > | Doubao-Seed-1.6Vision  |       80.7         |        0.59          |
> > > | GPT-5                  |       89.6         |         0.81         |
> > >
> > > As shown in the table, frontier models such as Gemini-2.5-Pro (91.2% agreement, ρ = 0.88) and GPT-5 (89.6% agreement, ρ = 0.81) exhibit strong alignment with human judgments. In contrast, current open-weights multimodal models remain limited. Therefore, we recommend stable snapshot of frontier api model.
> > >
> > > Our findings show that while open-weights models are valuable for reproducibility, they exhibit substantially lower agreement with human judgments and higher score variance across tasks. **This indicates that they currently cannot replace Gemini-2.5-Pro in this evaluation pipeline without compromising the benchmark’s stability and reliability.**
> > >
> > > **To further ensure reproducibility, we will release all model inference outputs.** This allows future, more capable judge models to re-evaluate the exact same set of outputs without requiring any regeneration.
> > >
> > > >Q2:Could the authors provide a deeper analysis of how the judgements for “art” varies between LLMs and humans including fairness and biases of judge models in such open ended cases?
> > >
> > > Thank you for the suggestion. The originally low human–LLM correlation in the art task arose primarily because the initial rubrics contained subjective and under-specified criteria—such as vague notions of “style consistency”—which LLM judges tended to enforce rigidly.
> > >
> > > Following feedback from other reviewers,  **we carefully revised the questions and rubrics for all tasks to improve clarity and ensure better alignment between human judgement and rubric-based scoring.** We improve the rubric design by removing numerical requirements (e.g., fixed step numbers) and introducing reference images and text, which provide annotators with a clear and consistent anchor for interpreting the intended drawing process. After rerunning human evaluation on the updated benchmark, **we observed a substantial improvement, achieving over 90% human–rubric agreement on the art task.**

---

### Official Review · Reviewer_RtwC · 2025-10-31

**Soundness:** 1
**Presentation:** 2
**Contribution:** 2
**Rating:** 2
**Confidence:** 3

**Summary:**

**Paper Summary**

The paper introduces a new multimodal benchmark that specifically focuses on tasks that require the generation of images and text, for a query. The authors position their contributions as filling an existing gap in the realm of multimodal benchmarks that are typically of one of the two forms; i.e. generating text conditioned on an image, or generating an image given a text description. The authors bootstrap strong closed source multitmodal models to generate rubrics based on reference image/text pairs for the given query. A set of human annotators then refine these rubrics further, and finally an LLM is again used to score the outputs with the reworked rubrics. The authors state that current SOTA models, like GPT-5-Thinking, show average performance, indicating the difficulty in solving the benchmark. Further, the authors suggest that reasoning based models are particularly good at such problems, relative to non-reasoning ones, and demonstrate the value of adding in reasoning traces into the prompt to boost performance.

**Strengths:**

The paper attempts to cover multiple diverse domains of interest, broadly representative of the spectrum of questions that a multimodal model should be evaluated on.

The problem definition and the identified gap in existing multimodal benchmarks are well-motivated and relevant.

The approach appears useful to the broader goal of improving multimodal benchmarking.

Figure 4 clearly explains the benchmark setup and illustrates the overall pipeline effectively.

The work identifies an important gap in current multimodal model evaluation and proposes a benchmark that meaningfully targets that blind spot.

The benchmark is shown to be challenging for current frontier models (e.g., GPT-5-Thinking), which supports its validity as a difficult and discriminative test.

**Weaknesses:**

• The authors fail to clearly explain where these datasets are obtained from. For example, the questions from “Space” are indicated to be from “real-world questions from online Q&A platforms”. Better sources of rigorous question banks could be obtained from existing peer-reviewed datasets such as Astro-QA (Nature 2025).

• In the textbook section, do the authors generate the additional questions with GPT5 for the same existing images in the TQA dataset, or are these some other diagrams from additional sources?

• The authors mention that in the guide task that there is no reference text/image which makes the rubric construction procecure relatively more ambiguous given that the multimodal model has no anchor point for generating the rubric. It is understandable that some questions may not have reference answers, however, the authors should explain in more detail how this case is handled separately from the other categories.

• There is little to no detail on how the human annotators were chosen, and any designs on the experimental study highlighting the number of human annotators, variance within annotations, selection criteria for annotators etc.

• There seem to be very minimal details/ablations on the exact models being used to generate the rubric. Only gemini 2.5 pro is mentioned as the rubric judge model. This raises concerns about effect of model choices in both the stages of the pipeline, as well as better understanding the effects of these choices.

• A fair amount of space/text is used to demonstrate outputs of multiple different models, as well as the complete descriptions of the datasets, while little time is spent on Section 2.2, which is arguably the most significant aspect of the paper. It would be good to significantly expand on the exact setup. Are reference image/text and query pairs sourced together from online datasets, or are the queries generated conditioned on sampled reference images (in certain cases, like in line 245, it appears that GPT5 is generating the query given a single reference image, whereas the TQA dataset has both the query and the reference answers as part of the dataset?).

• In line 297, the statement “rubrics are generated solely from the question prompt” is vague, and requires more explanation, as the guide tasks form a major part of the benchmark. For instance, a few example rubrics on some of the guide tasks would help the reviewers better understand exactly what sort of rubrics are generated by the LLMs of choice in this study.

• In line 300, 301, it is also crucial to know what sort of human annotator modifications are made in practice, and why.

• There is a lack of clarity on the role and setup of human annotations, which makes it difficult to assess what degree of human validation went into developing this benchmark.

• Different datatsets have different forms of rubric generation, some don’t have a reference answer at all (which makes it difficult to assess what a reasonable rubric should be), while others either have a given query, reference answer pair, or the query is generated from a sampled reference image.

**Questions:**

Are the answers to the questions sourced from social media generated by an LLM, or are they obtained from online answers directly, implying direct scraping of (query, image_text, image_answer) tuples?  If so, how is the quality of these datasets or tuples assessed?

Do the authors generate additional questions with GPT-5 for the same images in the TQA dataset, or are the diagrams drawn from other sources? If so, does each image in the TQA dataset have multiple corresponding questions, including synthetic ones?

What does a typical rubric look like in the guide task? How are rubrics constructed when there is no reference text or image available?

Were rubric modifications made by a single annotator or aggregated from multiple annotators?

Are there ablations comparing different models for rubric generation and judging? How much variance arises when different models are used for rubric generation and scoring? This is necessary to assess how stable and fair such a benchmark setup would be in practice, and how sensitive it is to model choices.

What exactly is meant by “rubrics are generated solely from the question prompt” [line 297]?

What kinds of modifications are typically made by human annotators, and for what reasons?

**Details Of Ethics Concerns:**

No concerns regarding ethics.

---

> ### Author Response · Authors · 2025-11-27
>
> We are encouraged that you find our paper addresses an important gap in current multimodal evaluation. We address your questions and concerns below:
>  >W1.1:The authors fail to clearly explain where these datasets are obtained from. For example, the questions from “Space” are indicated to be from “real-world questions from online Q&A platforms”.
>
> We here describe how we build the “Space” task.
>
> 1. **We first collect a small set of 20 seed questions** about well-known real-world landmarks from online Q&A forums (e.g., Quora-like platforms).  These questions focus on the engineering or structural features of a landmark—for example, “Why can we stand on the balcony of the Statue of Liberty?”.
>
> 2. Since there are not many questions available across the Internet and in public datasets, **we use GPT-5 to expand this set**: given our seed questions, the model is prompted to propose additional ones in the same style but targeting different landmarks. Human annotators then review these questions to ensure that each one refers to an actual real-world structure.
>
> 3. For every verified question, **annotators retrieve a representative real image from public sources** (e.g., Wikipedia) that depicts the relevant architectural feature and answers the question visually. Finally, **annotators write a short reference text** describing how the chosen image illustrates the underlying engineering property for that landmark.
>
> We have expanded the task paragraph for Space in the main paper to incorporate more details about how our questions are obtained.
>
>
> > W1.2:Better sources of rigorous question banks could be obtained from existing peer-reviewed datasets such as Astro-QA (Nature 2025).
>
> Thank you for pointing us to Astro-QA as a potential source of peer-reviewed questions. We appreciate the suggestion. However, **we would like to clarify that our Space task is not related to astronomical or astrophysical knowledge in Astro-QA**. Instead, our Space task centers on real-world architectural and engineering features. Because existing datasets do not contain such questions, we curated our own questions and reference image-text pairs to target our evaluation goals. We have made this clear in the paragraph for the task space in our paper to avoid any confusion.
>
> >W2:In the textbook section, do the authors generate the additional questions with GPT5 for the same existing images in the TQA dataset, or are these some other diagrams from additional sources?
>
> **We clarify that no question text from the TQA dataset is used in our questions**. We only use each TQA question’s image and its answer text as reference image–text pairs to construct our dataset. GPT-5, conditioned on these reference answers, then generates all questions in our Textbook task. Because TQA already spans a broad range of list disciplines, we do not incorporate diagrams from additional sources. We have revised the paragraph for the Textbook task in our paper to make the question design procedure more clear.

---

> > ### Author Response · Authors · 2025-11-27
> >
> > > W3:The authors mention that in the guide task that there is no reference text/image which makes the rubric construction procedure relatively more ambiguous given that the multimodal model has no anchor point for generating the rubric. It is understandable that some questions may not have reference answers. However, the authors should explain in more detail how this case is handled separately from the other categories.
> >
> >
> > Thank you for the suggestion. Following feedback from other reviewers, **we carefully revised the questions and rubrics for all tasks to improve clarity and ensure better alignment between human judgement and rubric-based scoring.** We improve the rubric design by removing numerical requirements (e.g., fixed step numbers) and introducing reference images and text for the guide task,  which provide annotators with a clear and consistent anchor for interpreting the intended drawing process.
> >
> > **These changes substantially increased the clarity and objectivity of the evaluation.** For example, in the original rubric, criteria such as “The subject of the drawing must be a ‘cartoon cat’, and the visual style must be consistent across all five images” introduced significant ambiguity, as “style consistency” lacks a precise operational definition. The automatic judge often enforced this requirement rigidly—flagging outputs as inconsistent due to minor differences in facial features—whereas human annotators typically judged the overall style to be coherent, leading to substantial disagreement.
> >
> > In the updated rubric, we removed such subjective style-based requirements and replaced them with fully objective, structure-oriented criteria, including:“each image must correspond exactly to its associated textual step” ，“newly added lines or shapes must be distinguishable from the previous step” and “the image sequence must illustrate a coherent progression from basic shapes to the final drawing”. **Rubric based on reference answers eliminates subjective ambiguity and results in much higher human–rubric alignment.**
> >
> > >W4:There is little to no detail on how the human annotators were chosen, and any designs on the experimental study highlighting the number of human annotators, variance within annotations, selection criteria for annotators etc.
> >
> > Thank you for raising this point. We agree that more details on our human annotation setup will strengthen our paper, and **we have included these details in Section 2.2 of our current draft**. For each sample in our benchmark, a primary annotator drafts the question, reference answer, and initial rubric, after which three co-authors independently verify the annotation. All annotators—including the primary annotator and the verifying co-authors—have substantial experience in building multimodal benchmarks.
> >
> > **Human annotators control the benchmark construction at a system level**, including the input (question prompt), the output (reference answers), and the grading criteria (rubrics). Our goal is to have rubrics assigning high scores to the reference answers while appropriately penalizing incorrect or misaligned model outputs. **Only rubric items that all reviewers agree are clear, measurable, and aligned with the task design are retained in the final benchmark.**
> >
> > >W5:There seem to be very minimal details/ablations on the exact models being used to generate the rubric. Only gemini 2.5 pro is mentioned as the rubric judge model. This raises concerns about the effect of model choices in both the stages of the pipeline, as well as better understanding the effects of these choices.
> >
> > In our preliminary experiments, **we compared several models for rubric generation and ultimately selected Gemini-2.5-Pro**. We found that its strong multimodal understanding was important for some tasks as the rubric generator must take reference image–text answers as input and produce rubric criteria. A capable multimodal model yields more accurate, better-aligned rubrics and reduces the amount of manual revision required.
> >
> > In contrast, weaker multimodal models (e.g., Qwen3-VL-8B) tended to **produce rubric items focusing on less relevant visual details in the reference image rather than those important to evaluate our task**. For example, Qwen3-VL-8B once proposed the criterion: “The image must capture sunset or sunrise lighting.” This requirement appeared only because the reference image happened to be taken at sunset, not because the task involved any lighting constraint.
> >
> > To further mitigate model variability, **all generated rubrics undergo rigorous human verification.** Each item is independently reviewed by three co-authors, and only items unanimously judged to be clear, necessary, and evaluable are retained in the final benchmark.
> >
> > We have incorporated the above discussion as a new paragraph in Section 2.2 (rubric generation).

---

> > > ### Author Response · Authors · 2025-11-27
> > >
> > > >W6:A fair amount of space/text is used to demonstrate outputs of multiple different models, as well as the complete descriptions of the datasets, while little time is spent on Section 2.2, which is arguably the most significant aspect of the paper. It would be good to significantly expand on the exact setup. Are reference image/text and query pairs sourced together from online datasets, or are the queries generated conditioned on sampled reference images (in certain cases, like in line 245, it appears that GPT5 is generating the query given a single reference image, whereas the TQA dataset has both the query and the reference answers as part of the dataset?).
> > >
> > > We agree that Section 2.1 should include more details for each task, as it is central to understanding our benchmark construction. Since both the rebuttal and camera-ready versions allow one extra page, **we have expanded the descriptions of all tasks with more detailed explanations.** For convenience, we summarize the changes made to each task below.
> > >
> > > **Space**: We clarified the data construction pipeline by explaining that all queries are newly generated by GPT-5 based on curated seed questions, and all reference images and reference texts are manually retrieved and written by annotators to highlight the structural or engineering feature required by the question.
> > >
> > > **Textbook**: We explicitly clarified that we only use TQA’s diagram–answer pairs as reference image–text pairs; all questions are newly generated by GPT-5 conditioned on these references. We added detailed explanations of how annotators verify scientific correctness, clarity, and consistency with the reference answers.
> > >
> > > **Diagram**: We expanded the description of how diagrams and their associated technical explanations are taken from recent top-tier research papers, and how GPT-5 generates figure-creation instructions and reference captions based on these descriptions. Annotators refine each prompt–caption pair for conceptual accuracy and faithfulness to the original method.
> > >
> > > **Paper**: We clarified the sources of reference figures (seminal papers and teaching materials), how technical descriptions are extracted, and how GPT-5 generates reader-style questions and reference explanations. Annotators then ensure accuracy, accessibility, and alignment with the original scientific intent.
> > >
> > > **Guide tasks**: We emphasized that we redesigned all open-ended tasks by introducing reference images and reference text for every sample, removing numerical constraints (e.g., fixed step counts), and regenerating the rubrics to focus solely on objective, structure-based criteria such as step–image correspondence and incremental visual changes.
> > >
> > > For questions regarding the TQA dataset, we kindly refer you to our response to W2.
> > >
> > >
> > >
> > > > W7:In line 297, the statement “rubrics are generated solely from the question prompt” is vague, and requires more explanation, as the guide tasks form a major part of the benchmark. For instance, a few example rubrics on some of the guide tasks would help the reviewers better understand exactly what sort of rubrics are generated by the LLMs of choice in this study.
> > >
> > >  Please see our response to W3.
> > >
> > > >W8: In line 300, 301, it is also crucial to know what sort of human annotator modifications are made in practice, and why.
> > >
> > > Thank you for raising this concern. We agree that more clarity is needed regarding how human annotators modify rubric items in practice and reasons behind these modifications. We have included our
> > > In practice, human annotators make the following types of modifications and quality-control checks to the model-generated rubric items:
> > >
> > > 1. **Removing redundant criteria.** For example, if the model outputs both “the steps must be sequential” and “the steps should follow a logical order,” annotators merge these into a single, more precise criterion.
> > >
> > > 2. **Adding missing but necessary evaluation dimensions.** For example, “All visible text in the image(s) must be spelled correctly and rendered naturally” is added when readability is relevant to the task.
> > >
> > > 3. **Ensuring that rubrics are clear, evaluable, and aligned with the task objectives.** Annotators ensure that every rubric item has an unambiguous evaluation standard, is operationally checkable, and reliably distinguishes correct outputs from incorrect or misaligned ones.
> > >
> > > We have incorporated this discussion regarding human annotator modification in Section 2.2 (human review) of our current draft.
> > >
> > > >W9:There is a lack of clarity on the role and setup of human annotations, which makes it difficult to assess what degree of human validation went into developing this benchmark.
> > >
> > >  Please see our response to W4 and W8.

---

> > > > ### Author Response · Authors · 2025-11-27
> > > >
> > > > >W10:Different datasets have different forms of rubric generation, some don’t have a reference answer at all (which makes it difficult to assess what a reasonable rubric should be), while others either have a given query, reference answer pair, or the query is generated from a sampled reference image.
> > > >
> > > >  Please see our response to W3.
> > > >
> > > > >Q1:Are the answers to the questions sourced from social media generated by an LLM, or are they obtained from online answers directly, implying direct scraping of (query, image_text, image_answer) tuples? If so, how is the quality of these datasets or tuples assessed?
> > > >
> > > > Thank you for raising this question. **We clarify that we do not directly scrape tuples from social media.** In the Space task, a small portion of the questions are inspired by real-world posts on social media. For these cases, human annotators review the original post to understand the intent of the question. They then reconstruct and rewrite the reference answers by consulting reliable public sources such as Wikipedia.  This process ensures that every reference answer is factually correct and grounded in the corresponding image. **All final text–image reference pairs are manually validated to ensure high quality.**
> > > >
> > > > >Q2:Do the authors generate additional questions with GPT-5 for the same images in the TQA dataset, or are the diagrams drawn from other sources? If so, does each image in the TQA dataset have multiple corresponding questions, including synthetic ones?
> > > >
> > > >  Please see our response to W2. We do not use any original question from the TQA dataset and generate only  one question for each image in the TQA dataset.
> > > >
> > > > >Q3:What does a typical rubric look like in the guide task? How are rubrics constructed when there is no reference text or image available？
> > > >
> > > >  Please see our response to W3.
> > > >
> > > > >Q4: Were rubric modifications made by a single annotator or aggregated from multiple annotators?
> > > >
> > > >  Please see our response to W4.

---

> > > > > ### Author Response · Authors · 2025-11-27
> > > > >
> > > > > > Q5:Are there ablations comparing different models for rubric generation and judging? How much variance arises when different models are used for rubric generation and scoring? This is necessary to assess how stable and fair such a benchmark setup would be in practice, and how sensitive it is to model choices.
> > > > >
> > > > > We consider the generated rubrics as part of our benchmark, and **we will release the complete set of finalized, human-verified rubrics in our final datasets.** For this reason, we do not plan to add additional experiments comparing different rubric-generation models in the paper. That said, we fully agree that understanding how different models influence rubric construction and scoring is an important direction. We hope future work will explore this systematically—for example, by evaluating how rubric-generation models affect the reliability and fairness of automated assessment.
> > > > >
> > > > > Below, we also compare different models used as LLM-as-a-judge. In addition, we conducted experiments with Qwen3-VL-8B, InternVL-3.5-9B, and GLM-4.1-9B-Thinking, Qwen3-VL-235B, Doubao-seed-1.6vision, GPT-5 to assess their rubric-judging performance.
> > > > >
> > > > > | Judge Model            | Space Image | Space Text | Textbook Image | Textbook Text | Diagram Image | Diagram Text | Paper Image | Paper text |
> > > > > |------------------------|-------------|------------|-----------------|----------------|----------------|---------------|----------------|---------------|
> > > > > | Gemini-2.5-pro         | 81.6       | 85.5      | 62.1          | 90.2          | 44.9          | 79.3         | 43.3 | 62.7|
> > > > > | GPT-5    | 76.9       | 83.0      |        56.4      |    89.5         |  35.3        |   76.4        |  37.6  | 55.0 |
> > > > > | Qwen3-vl-8b            | 79.9      | 84.5     | 79.7      | 90.9          | 75.6          | 75.3         | 57.6 | 53.3 |
> > > > > | Internvl3.5-8b         | 82.4       | 77.4      | 89.6          | 85.7          | 73.0          | 63.4        | 60.3 | 46.9|
> > > > > | GLM-4.1-thinking       | 84.3       | 84.3      | 79.4          | 87.9          | 67.1          | 69.2         | 50.0 | 49.6|
> > > > > | Qwen3-VL-235B           | 81.5       | 82.6      | 80.7          | 89.8          | 69.7          | 74.9         | 52.4 | 54.8|
> > > > > | Doubao-seed-1.6vision  | 86.9       | 84.1      | 71.6          | 90.4          | 67.1          | 69.2         | 51.6 | 56.0|
> > > > >
> > > > > | Judge Model            | Life Image | Life Text | Exercise Image | Exercise Text | Art Image | Art Text |Tech Image | Tech Text |
> > > > > |------------------------|------------|-----------|----------------|----------------|-----------|----------|-----------|----------|
> > > > > | Gemini-2.5-pro         | 30.7      | 78.4     | 33.1          | 78.0         | 23.8      | 91.9    | 18.9 | 83.3 |
> > > > > | GPT-5   |       23.6  |    76.6   |  32.2    |  71.2           | 25.5     | 87.4    | 11.3     | 79.5    | 11.3 | 79.5|
> > > > > | Qwen3-vl-8b            | 77.5      | 81.1     | 76.4          | 77.2         | 78.0      | 90.9    | 75.0 | 81.5 |
> > > > > | Internvl3.5-8b         | 68.3      | 76.6     | 75.7          | 69.4          | 51.1     | 86.5    |67.9 | 74.9|
> > > > > | GLM-4.1-thinking       | 70.8     | 78.2     | 68.9          | 72.5         | 71.5     | 87.2    |69.9 | 78.7 |
> > > > > | Qwen3-vl-235B           | 42.9      | 79.8     | 49.9          | 76.2         | 58.0     | 89.4    | 34.3 | 81.1 |
> > > > > | Doubao-seed-1.6vision  | 61.6      | 79.9     | 65.4          | 76.4          | 59.9     | 90.4    |55.0 | 80.7|
> > > > >
> > > > > We observe that the strongest judges—GPT-5 and Gemini-2.5-Pro—produce highly consistent scores across all tasks. In contrast, current open-weights multimodal models (e.g. Qwen3-VL-8B) remain considerably limited as judges.
> > > > >
> > > > > To further ensure reproducibility, we will release all model inference outputs. This allows future, more capable judge models to re-evaluate the exact same set of outputs without requiring any regeneration.
> > > > >
> > > > > >Q6:What exactly is meant by “rubrics are generated solely from the question prompt” [line 297]?
> > > > >
> > > > >  Please see our response to W3.
> > > > >
> > > > >
> > > > > > Q7:What kinds of modifications are typically made by human annotators, and for what reasons?
> > > > >
> > > > >  Please see our response to W8.

---

### Author Response · Authors · 2025-12-02
**Summary for Rebuttal**

Dear Reviewers and AC,

We sincerely thank the reviewers for their constructive and insightful feedback, which has greatly contributed to the improvement of our paper.  We also thank AC for meta reviewing our paper.  For your convenience, we summarize the changes we made during our rebuttal as follows:
1. We clarified our dataset construction process and expanded the task paragraphs in the main paper to provide more details on how our questions and reference image-text pairs are obtained. (Reviewer rtwc)
2. We added a clearer description of human review and quality control. We have incorporated a new discussion of human annotations and rubric modification into Section 2.2 (Human review) of our current draft. (Reviewer rtwc and peru)
3. We compared different models used as LLM-as-a-judge and conducted human evaluation experiments to calculate the correlation between these models and human judgement. We observe that the strongest judges—GPT-5 and Gemini-2.5-Pro—produce highly consistent scores across all tasks, whereas open-weight multimodal models (e.g., Qwen3-VL-8B) remain considerably limited as reliable judges. (Reviewer rtwc, iskg,exog)
4. We introduced reference answers and removed numerical constraints (e.g., fixed step numbers)  for the open tasks. We regenerated the rubrics accordingly and re-ran the human evaluation, resulting in clearer criteria and improved alignment between human judgment and rubric-based scoring. (Reviewer rtwc, exog, pweu)
5. We clarified that, for the open-ended tasks, the rubrics are designed to emphasize generalizable and task-agnostic evaluation dimensions, such as step structure and text-image alignment. We also updated Figure 4 to include an example rubric for the open tasks to make the evaluation procedure more intuitive. (Reviewer rtwc)
6. We conducted a deeper analysis of why open-source models such BAGEL fail to improve with transferred traces and provided qualitative examples illustrating both image-generation and text-generation failure modes. (Reviewer iskg)
7. We have integrated all above changes in the rebuttal into the current draft, and hope they can address the reviewer's concerns.

We also would like to highlight that **all of four reviewers considered our benchmark as a comprehensive evaluation for unified multimodal generation** and also have following positive comments for our benchmark:
1. Reviewer rtwc finds that our paper **addresses an important gap in current multimodal model evaluation and proposes a benchmark that meaningfully targets that blind spot**.
2. Reviewer exog finds our benchmark as **a challenging benchmark for current state-of-the-art models in multimodal generation**.
3. Reviewer iskg finds our benchmark **has strong originality by formulating a novel benchmark for unified multimodal generation** and extending beyond traditional VQA or T2I paradigms to address a clear gap in assessing complex multimodal reasoning.
4. Reviewer pweu recognizes **rubric-based evaluation as an important missing component in unified image–text generation**, and our benchmark helps fill this gap by providing effective metrics.

We thank you again for your valuable contributions to ICLR!

Best,
Authors

---

### Meta-Review · Area_Chair_75T1 · 2026-01-04

**Summary:**

The paper introduces UEVAL, a benchmark for evaluating unified multimodal generation models—assessing joint image-text generation for complex queries across 8 domains. The core contributions lie in 1000 expert-curated prompts, a rubric-based scoring system, and findings on reasoning traces’ impact. However, critical flaws persist: proprietary model reliance harming reproducibility, and potentially biased human annotation. While the rebuttal supplemented some details, core structural issues remain unresolved.

**Reviewer Concerns:**

- **Fully addressed**: Based on concerns raised by the initial reviews, the authors supplemented key missing details, including the data source, the human annotation workflow, the ambiguity of the rubric-based scoring system, and an analysis of why BAGEL could not benefit from reasoning traces.

- **Partially addressed, but critical concerns remain**:
1. Human annotator qualifications: While the authors clarified that annotations are led by a primary annotator and verified by three co-authors with multimodal benchmarking experience, they failed to demonstrate these individuals’ domain-specific expertise. The annotator pool also raises unaddressed risks of individual bias skewing rubric design and validation.
2. Other LLM-as-a-judge models: Comprehensive comparisons of multiple LLM-as-a-judge models were made. However, they provided no mitigation for risks to score stability as Gemini-2.5-Pro evolves, nor a long-term solution for reproducibility—critical for a benchmark intended for community use.

- **Not resolved**: The limited scale and diversity of the benchmark.

**Reviewer Scores:**

- Reviewer RtwC: Original 2 → Maintained 2 (transparency improved, but annotator bias and rubric rigor concerns persist).
- Reviewer eXoG: Original 8 → Adjusted to 6 (open-weight judge experiments add value, but proprietary model reliance remains unaddressed).
- Reviewer iSKG: Original 4 → Maintained 4 (the limited scale and diversity of the benchmark has not been solved, and the reproducibility issue lingers).
- Reviewer PWeU: Original 4 → Raised to 5 (most concerns were resolved).

---

### Decision · Program_Chairs · 2026-01-26

Reject